# On the importance of statistics in molecular simulations for thermodynamics, kinetics and simulation box size

Vytautas Gapsys, Bert L de Groot*

Computational Biomolecular Dynamics Group, Max-Planck Institute for Biophysical Chemistry, Göttingen, Germany

**Abstract** Computational simulations, akin to wetlab experimentation, are subject to statistical fluctuations. Assessing the magnitude of these fluctuations, that is, assigning uncertainties to the computed results, is of critical importance to drawing statistically reliable conclusions. Here, we use a simulation box size as an independent variable, to demonstrate how crucial it is to gather sufficient amounts of data before drawing any conclusions about the potential thermodynamic and kinetic effects. In various systems, ranging from solvation free energies to protein conformational transition rates, we showcase how the proposed simulation box size effect disappears with increased sampling. This indicates that, if at all, the simulation box size only minimally affects both the thermodynamics and kinetics of the type of biomolecular systems presented in this work.

*For correspondence:
bgroot@gwdg.de

## Introduction

Molecular simulations sample well defined thermodynamic ensembles, thus providing a representation of the physical world in silico. Naturally, the simulated reality is limited in accuracy by the underlying assumptions and approximations (*van Gunsteren and Berendsen, 1990*). For example, a frequently noted shortcoming of molecular dynamics (MD) simulations is its dependence on the force field, that is, simplified representation of the electronic ground state potential energy. Large efforts are continuously dedicated to improve force field accuracy (*Lindorff-Larsen et al., 2012*).

Another major simulation accuracy determining factor is the sampling convergence. For the cases where the phase space is not thoroughly explored, e.g. relevant protein conformations are not sampled, the subsequently estimated thermodynamic and kinetic properties will likely have a substantial associated error. A failure to properly assess this error may be critical, leading to an erroneous interpretation of the simulation data and a wrong overall conclusion (*Knapp et al., 2018*). Considering the stochastic nature of common sampling algorithms such as molecular dynamics simulations, biomolecular trajectories represent a multidimensional random walk of which the analysis is especially prone to suffer from sampling deficiencies (*Hess, 2002*).

Recently, reports have appeared on a possible effect of the simulation box size on thermodynamic quantities in atomistic molecular dynamics simulations (*El Hage et al., 2018*; *El Hage et al., 2019*; *Asthagiri and Tomar, 2020*). Specifically, the role of solvation and a box-size-dependent hydrophobic effect have been claimed as the underlying physical cause. However, these studies have been challenged, showing no significant box size effects when a larger statistical sample was studied (*Gapsys and de Groot, 2019*; *Mehra and Kepp, 2019*), indicating a lack of statistical significance of the original results that appeared to show a box size dependence. This scientific discussion makes the simulation box size variable an interesting candidate for further investigation in the light of statistical significance of the calculated thermodynamic and kinetic measures.

In the current work, we closely examine box size effects in different systems varying from the solvation free energy of a small molecule to the kinetics of a protein conformational change. We use a rigorous statistical framework to evaluate how the employed statistics affect the conclusions.

## Results

### Thermodynamics: Solvation of small molecules

As a first example, we have investigated the box size dependence of the computed hydration free energy of the small molecule anthracene by employing the alchemical approach. The alchemical method (*Straatsma and McCammon, 1992*) allows circumventing computationally expensive paths along the thermodynamic cycle, e.g. pulling a molecule from the gas phase into solvent, for the case of solvation free energy calculations. This is achieved by exploiting an alternative, alchemical path: for the hydration free energy example, the Hamiltonian of the molecule in gas phase is transformed into the Hamiltonian of the molecule coupled to water. Sampling this path is not feasible in nature, but is possible by computation. Since the free energy is a thermodynamic state variable, it does not depend on whether a physical or alchemical pathway has been used for calculation.

We computed the hydration free energy in boxes ranging from 473 to 5334 solvating water molecules. The smallest box size was chosen such as to minimally satisfy the simulation condition that the box size exceeds twice the non-bonded cut-off radius in all three spatial dimensions. Twenty repeats were included per box size to study the spread from individual estimates. As can be seen in *Figure 1B*, in contrast to *Asthagiri and Tomar, 2020*, no trend in the computed hydration free energy was observed as a function of the employed simulation box size, when all repeats (N = 20) are taken into account. Naturally, if we were to rely on single realizations of molecular dynamics trajectories, it is possible to obtain any type of trend suggesting a box size dependence: in the middle panel of *Figure 1B* we highlight an arbitrary selection of an upward, downward, as well as upward followed by downward trends. However, neither trend is statistically significant and merely illustrates the erroneous conclusion that may be drawn from anecdotal evidence. This analysis also clearly illustrates the importance of reporting uncertainty estimates for the calculated observables: depicting confidence intervals for the ΔG estimates (*Figure 1B* right panel) would help avoiding making unfounded claims about the depicted trends.

These findings are well in line with an earlier investigation demonstrating absence of any box size effects in calculating solvation free energies of small neutral molecules (*Parameswaran and Mobley, 2014*).

### Thermodynamics: $G_B$ protein

Although no box size effects have been observed when analyzing solvation free energies of small ligands, it cannot be excluded that, in case the box size dependence is a subtle phenomenon, it might manifest itself in calculating hydration $\Delta G$ for larger molecules, e.g. proteins. In fact, *Asthagiri and Tomar, 2020* have reported a box size dependence of a small 56 residue protein $G_B$ when computing solvation free energy by means of a quasichemical theory. Here, we also used protein $G_B$ as a model system (*Figure 2A*) to investigate whether the solvation free energy based on the molecular dynamics sampling depends on the size of the simulation box.

Since coupling a large molecule to solution imposes a major sampling challenge, we separated the whole hydration $\Delta G$ calculation into two independent steps. Firstly, we switch on the charges on the solute, this way evaluating the electrostatic contribution to the solvation free energy. In the second step, we introduce the van der Waals interactions of protein atoms, this way estimating the hydrophobic contribution of Pauli repulsion and attractive dispersion interactions. For both of these steps we also computed the respective $\Delta G$ of charge and van der Waals interaction introduction in vacuum.

Switching on protein charges in water (*Figure 2B* left panel) appears to be independent of the box size. The simulations without solvent (*Figure 2B* middle panel) exhibit strong box size dependence. The latter result of simulations in vacuum is easy to explain when taking into consideration the treatment of long range electrostatic interactions: in our simulations we followed the nowadays standard approach of Ewald summation (Particle-Mesh-Ewald, PME method [*Darden et al., 1993*; *York et al., 1993*]) to account for the electrostatic interactions in a system with periodic boundaries.

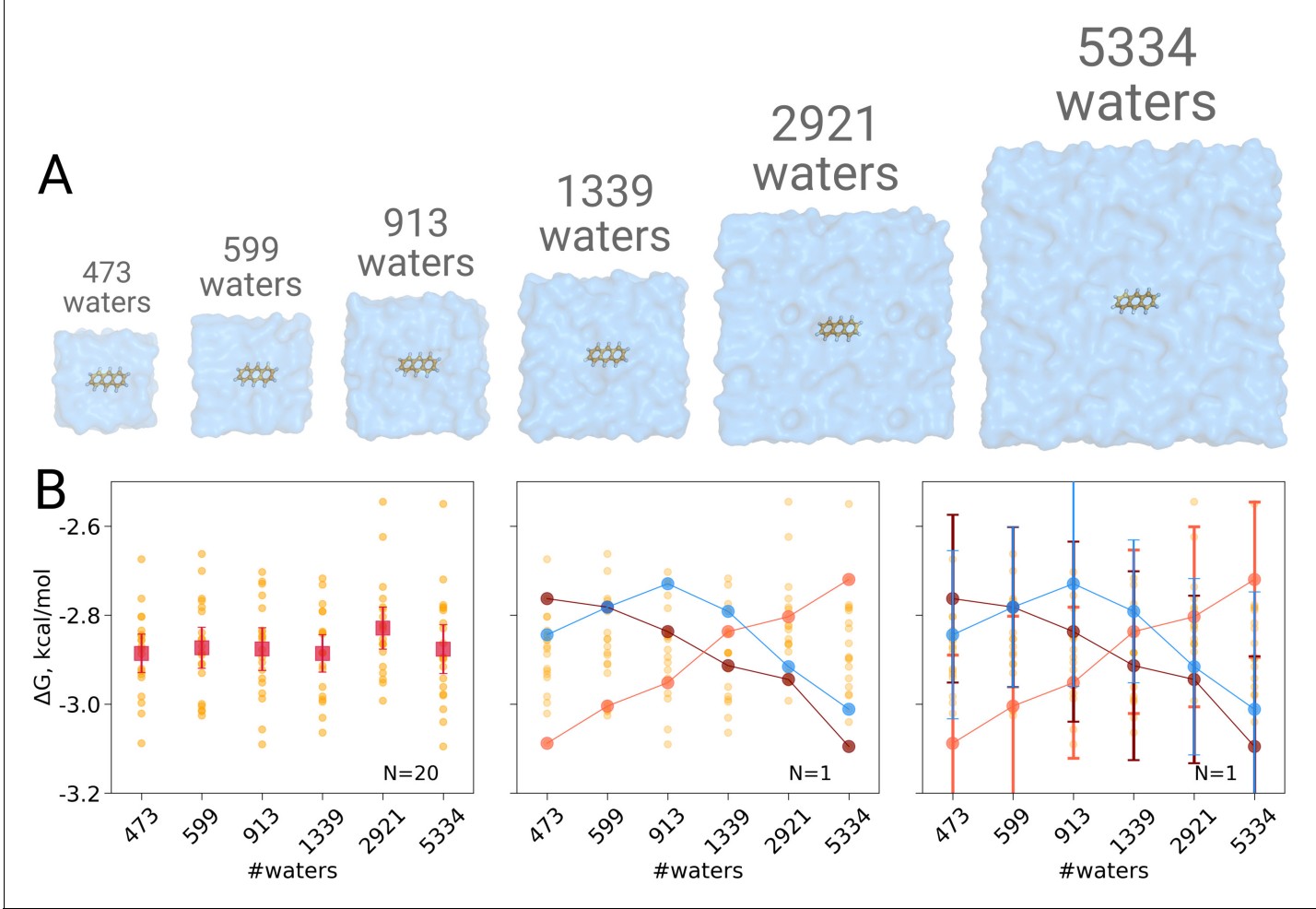

**Figure 1.** Solvation free energy of anthracene molecule. The simulations were performed in boxes of 6 different sizes (**A**). We repeated the calculation 20 times for each simulation box. The outcome of each calculation is depicted as an orange circle in panels **B** and **C**. (**B**) Averages over 20 simulations together with the associated 95% confidence intervals reveal that there is no box size dependence for the solvation free energy (left panel). Three unlikely, yet possible, scenarios are depicted in the middle panel, where results from only one calculation per simulation box are considered. Given this insufficient (N = 1) sampling of solvation free energies, we could reach a false conclusion that the $\Delta G$ values increase, decrease or increase and then decrease with a larger simulation box. The right panel shows the same trends as in the middle panel, but the estimates have their 95% confidence intervals depicted as well: considering the statistical uncertainty illustrates that all the trends in terms of box size dependence based on anecdotal evidence are not statistically significant.

Decreasing the box size in vacuum alters protein's environment, as the distances between the periodic images decrease.

In the solvated system, PME is used as well, but the electrostatic effect between the periodic images is screened by water molecules. Following best practices of the system setup for molecular dynamics simulations ensures a sufficiently large distance between the periodic images, such that the thermodynamic properties of the system remain box size independent. For example, in the current setup, the minimal distance between the protein surface and the box edge was at least 1 nm even in the smallest simulation box. Reducing the box size allowing ~0.5 nm distance between the solute and the box edge introduces artifacts where water screening is not sufficient and solvation shells of the periodic images can interact with each other. Such an extremely small box introduces a clear box size dependence (*Figure 3A*) and is not recommended for any MD simulation.

Another illustration of the effects of screening is depicted in *Figure 3B*. Scaling the charges on water molecules to retain only 10% of the original charge reduces solvent's screening strength making the solvent more similar to a Lennard-Jones fluid. In turn, the protein's charge introduction free energies become more similar to those in vacuum and a box size dependence emerges.

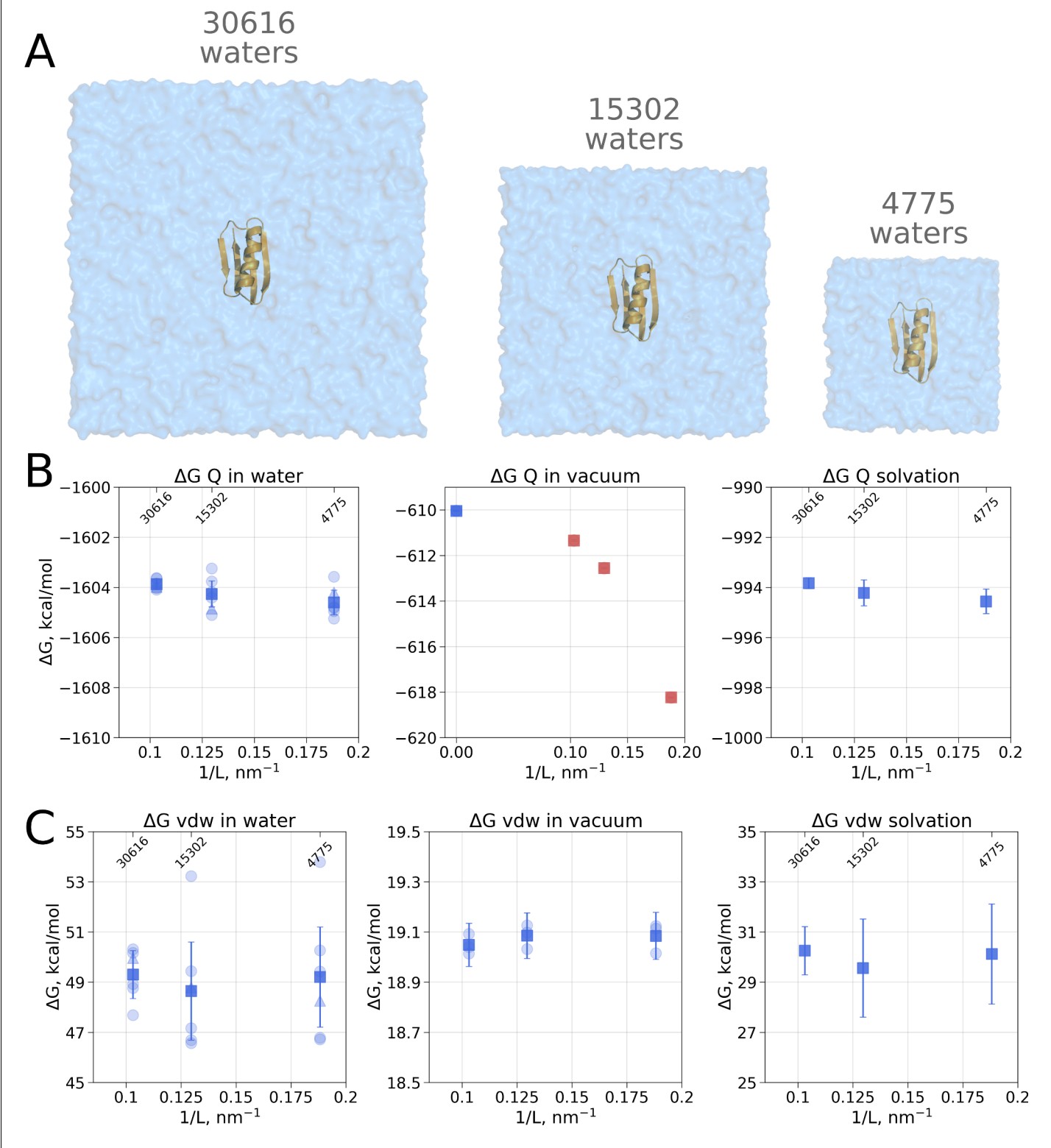

**Figure 2.** Contributions to the solvation free energy of the $G_B$ protein. The protein was simulated in boxes of 3 different sizes (**A**). (**B**) Contribution to solvation $\Delta G$ from charge introduction in water (left), vacuum (middle) and $\Delta G_{water} - \Delta G_{vacuum}^{aperiodic}$ (right). The latter term was calculated by considering a $\Delta G_{water}$ value (i.e. free energy of switching on protein charges in water) for each of the box sizes (left panel) and $\Delta G_{vacuum}$ (i.e. free energy of switching on protein charges in the gas phase) calculated in the infinite box without periodic boundaries (blue square in the middle panel). Blue markers denote those cases that are box size independent, while red symbols are for the box size-dependent contributions to $\Delta G$. (**C**) Contribution to solvation $\Delta G$ from

*Figure 2 continued on next page*

*Figure 2 continued*

switching on van der Waals interactions of the solute. In the panels (**B**) and (**C**) circles denote values obtained from individual free energy perturbation (FEP) calculations, triangle symbol marks those FEP runs where Hamiltonian replica exchange (HREX) was used, squares mark averages over the individual calculations. Upper x-axis marks the number of waters in the system.

Overall, to obtain the net electrostatic contribution to solvation free energy we can use the calculation in any of the sufficiently large solvated water boxes (*Figure 2B* left panel) and subtract the $\Delta G$ value calculated in an infinitely large non-periodic vacuum box (illustrated by the blue square in *Figure 2B* middle panel). This ensures that the electrostatic component of hydration free energy is independent of the box size. Also, it is worth noting that in our calculations of $\Delta G_Q$ we did not need to apply any additional corrections, while Asthagiri and Tomar used an analytical correction (*Hummer et al., 1998*) to remove box size dependence of this $\Delta G$ component. The need for such a correction might depend on the particular details how the MD software computes the solvation free

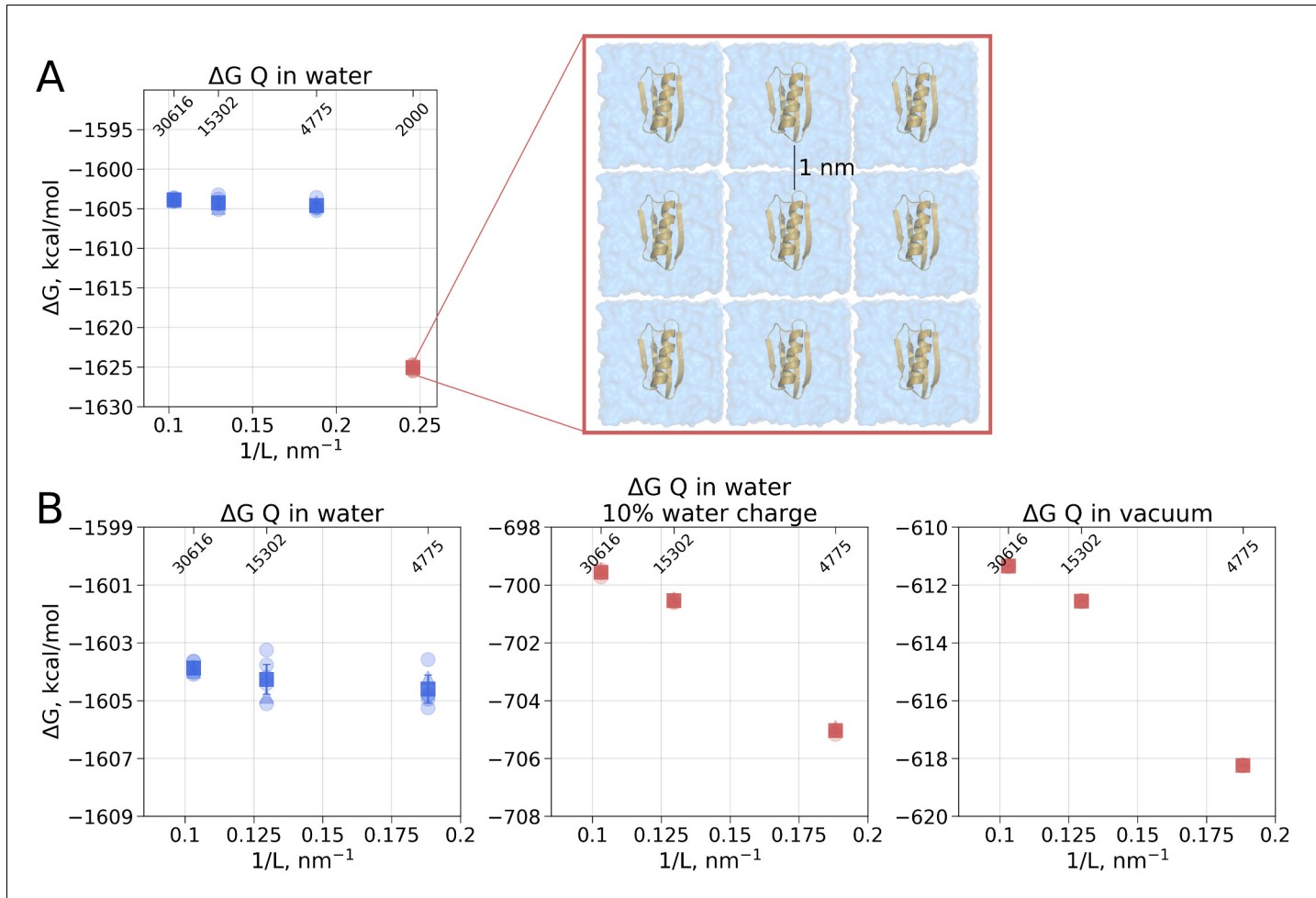

**Figure 3.** Illustration of artifacts introduced in the solvation $\Delta G$ calculation upon charge introduction. (**A**) Strongly reduced simulation box size has an effect on the calculated $\Delta G$ value. Protein interactions with its periodic image and interactions of the solvation shells of the solute have a significant impact. (**B**) The effect of electrostatic screening by water molecules. The values in the panel on the left were obtained with the regular TIP3P water and show no box size dependence. Once water screening is reduced by scaling water charges to 10% of their original value (middle panel), the estimated $\Delta G$ values start approaching those obtained in vacuum simulations (right panel). In (**A**) and (**B**) blue markers denote those cases that are box size independent, while red symbols are for the box size-dependent Circles denote values obtained from individual free energy perturbation (FEP) calculations, triangle symbol marks those FEP runs where Hamiltonian replica exchange (HREX) was used, squares mark averages over the individual calculations. Upper x-axis marks the number of waters in the system.

energy. Gromacs (*Abraham et al., 2015*), the simulation package used in this work, considers all the terms, including short and long range solute-solvent, solute-solute, as well as self interactions.

Another marked difference between the calculation reported by Asthagiri and Tomar and our work comes from the estimated absolute value of the $\Delta G_Q$ component: $-662$ kcal/mol and $-994$ kcal/mol, respectively. There are several possible reasons underlying such a large discrepancy. The calculation of the solvation free energy of a whole protein is a highly specific endeavour, thus the absolute value of the estimated $\Delta G$ will strongly depend on the particular details of the simulation setup. To facilitate convergence and avoid protein unfolding for the simulations in vacuum Asthagiri and Tomar chose to freeze all the degrees of freedom of the protein. In our setup we only restricted the protein motions by harmonic position restraints. This disparate solute treatment itself contributes to the final difference in $\Delta G$. An even larger contribution to the discrepancy can be expected from having all the protein degrees of freedom constrained (or restrained), as it exaggerates the influence of the starting structure for simulation: the internal dynamics of the solute is restricted and the solvent will interact with the initial conformer only.

Another aspect that we took particular care of was to properly equilibrate water molecules for the solvation free energy calculations. In the first step we performed annihilation of the van der Waals interactions of uncharged $G_B$ using Hamiltonian replica exchange simulations. This way the replicas of the protein were exchanged in 32 discrete steps between the pure water box and the water box with the protein coupled to the system via van der Waals interactions. In turn, the end state of the protein with the van der Waals interactions switched on was further used to initialize $\Delta G_Q$ calculations. Naturally, such an exhaustive equilibration allowed water molecules to explore deep pockets of the protein, later contributing to the final free energy estimate.

Lastly, a considerable amount of sampling is required to reach convergence for an alchemical solvation of the whole protein, as the phase space overlap between neighboring windows needs to be ensured. In this work we invested 1.38 μs of sampling for the $\Delta G_Q$ solvation in water for each box size including sampling enhancement by means of Hamiltonian replica exchange. Considering different sampling times in this work and that reported by Asthagiri and Tomar might further contribute to the discrepancy in the estimated $\Delta G_Q$ values.

As for the van der Waals interaction contribution to the solvation free energy, no box size dependence is present neither in the solvated, nor in the vacuum state (*Figure 2C*). This result shows that the box-size dependence for hydration free energies as reported by *Asthagiri and Tomar, 2020* is not a general phenomenon but a feature of the applied quasichemical theory.

It is of interest to note that both the anthracene and protein G van der Waals coupling examples address the hydration free energy of a hydrophobic molecule. Hence, these also serve to test the hypothesis of a possible box size dependence of the hydrophobic effect, as had been suggested (*El Hage et al., 2018*). Even though according to one theory of hydrophobicity the anthracene and protein G cases are relatively small systems (*Chandler, 2005*), the current series do not provide any evidence that the solvation-driven hydrophobic effect is significantly affected by solvation beyond the first solvation layers.

## Thermodynamics: alanine dipeptide

The systems analyzed the solvation $\Delta G$ calculations described in the previous sections had limited internal degrees of motions: anthracene is a rigid planar molecule, while $G_B$ protein was restrained during the simulations to facilitate convergence. To explore possible box size influence on the internal dynamics of biomolecules we determined free energy profiles of alanine dipeptide and dihydrofolate reductase.

Firstly, we studied the alanine dipeptide, a well established reference system in the field as a minimal model system for molecular conformational transitions. It undergoes spontaneous transitions between two major conformations on the nanosecond timescale, therefore rendering it an ideal model system to study both the kinetics as well as thermodynamics by computational methods. *Figure 4* shows the potential of mean force for four different simulation box sizes along the $\Psi$ dihedral angle that we use as a reaction coordinate to distinguish the two conformations. As before, we analyzed the results using only subsets of the data, to study sampling convergence. Also here, limited sampling, e.g. 0.1% of the data, may lead to erroneous conclusions about the box size-dependent ratio between the alanine dipeptide conformers. The discrepancies between the free energy profiles generated in different simulation boxes disappear with more sampling data included.

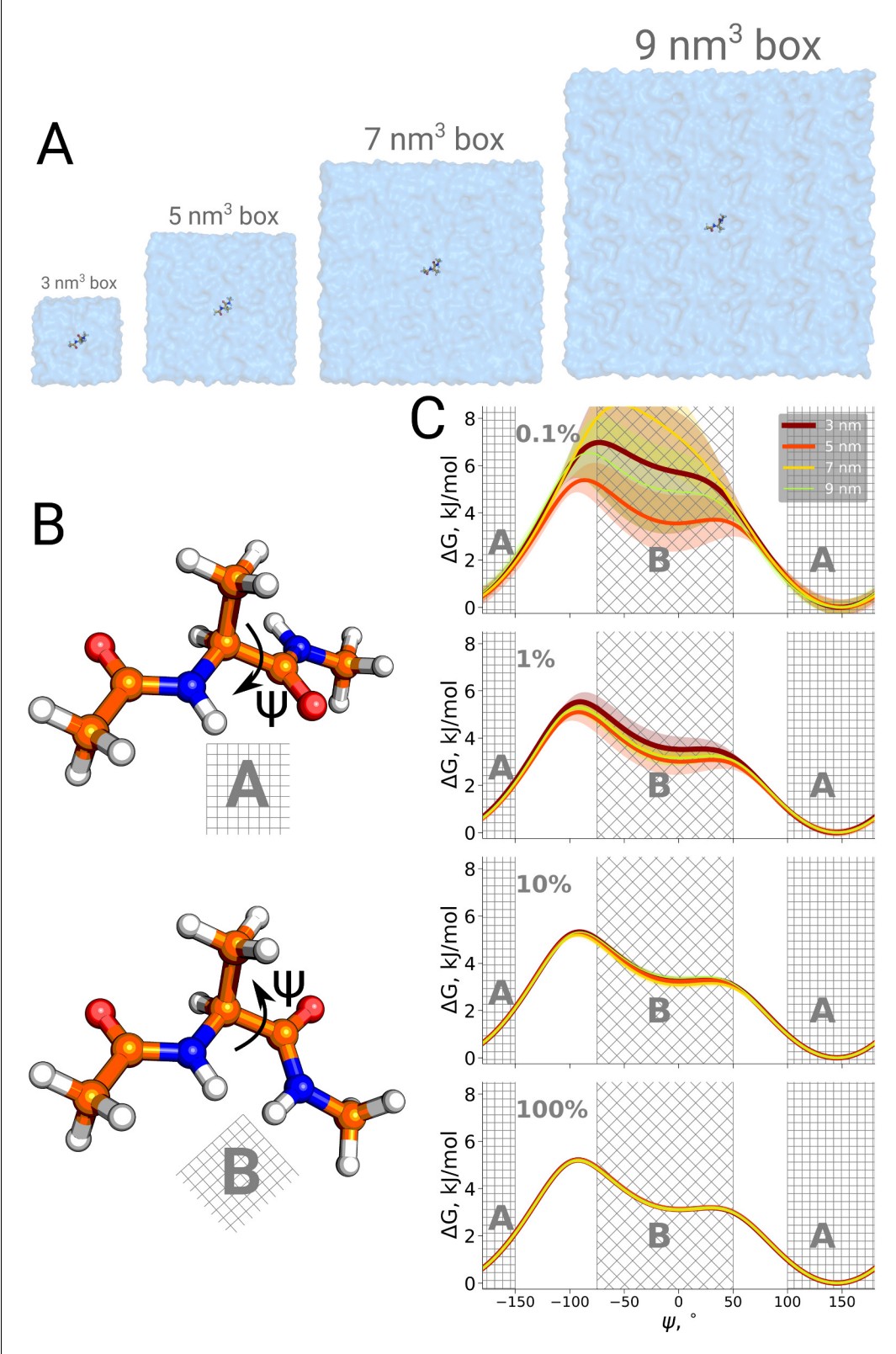

**Figure 4.** Potential of mean force for alanine dipeptide. Alanine dipeptide was simulated in boxes of 4 different sizes (A). (B) The molecule undergoes a well defined transition characterized by rotation around its Ψ dihedral backbone angle. (C) Free energy profiles along the reaction coordinate: the Ψ dihedral angle. The sub-panels going from top to bottom depict profiles constructed from an increasing amount of simulation data: 0.1%, 1%, 10% and 100% of the whole available data. Uncertainties are depicted by shaded areas and represent standard errors obtained from 10 independent repeats.

## Thermodynamics: dihydrofolate reductase

In the literature one can find a number of studies reporting on a box size effect on the population ratios of conformers of a simulated biomolecule (*Hünenberger and McCammon, 1999*; *Weber et al., 2000*; *Babu and Lim, 2020*). In some cases, these effects can be well rationalized: due to the limitations of computing power, the earlier studies relied on implicit solvation (*Hünenberger and McCammon, 1999*) or very short simulations without repetitions (*Weber et al., 2000*).

A recent study, however, draws attention as a box size effect has been reported from several independent repeats of an explicitly solvated protein dihydrofolate reductase. *Babu and Lim, 2020* reported a strongly box size-dependent potential of mean force (PMF) profile for the M20 loop motion (*Figure 5B*) of the protein. In their study, the protein was solvated in three cubic boxes of varying size (box edge length of 7, 8 and 9.3 nm). Even the smallest box was sufficient to avoid protein self interactions (minimal distance between periodic images at least 1.4 nm). In spite of this apparently sufficient solvation, the authors have reported the PMF profile in the smallest box to deviate significantly from the other two larger boxes. It is therefore interesting to investigate what might give rise to such an effect.

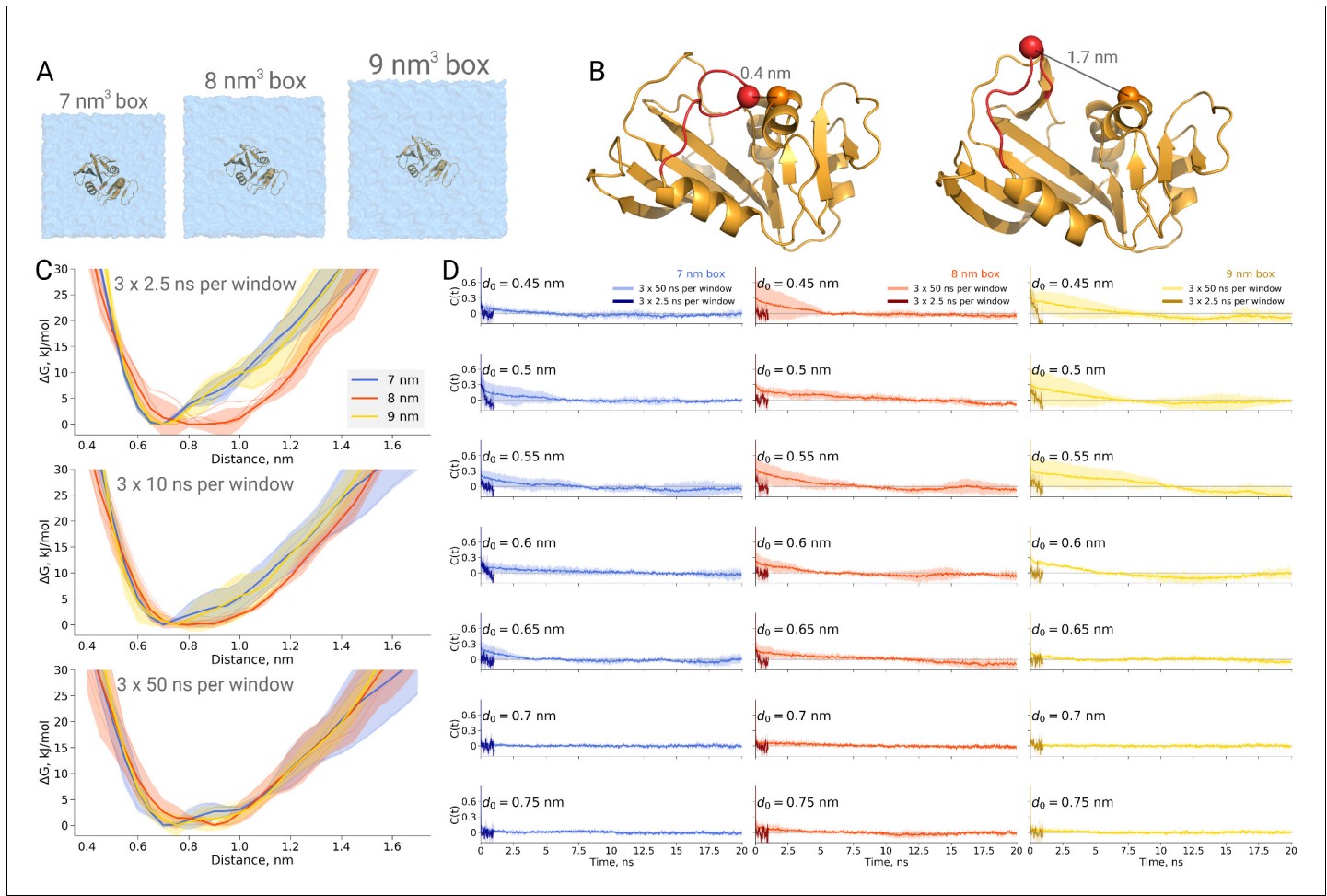

**Figure 5.** Thermodynamics of dihydrofolate reductase. Dihydrofolate reductase was simulated in boxes of 3 different sizes (**A**). (**B**) The distance between C$_\alpha$ atoms of residues 18 and 45 was used as a reaction coordinate. Here, loop M20 is coloured in red. (**C**) Free energy profiles along the reaction coordinate were constructed from three simulation repeats. Upper panel: 2.5 ns sampling time of which 0.5 ns were discarded for equilibration. Middle panel: 10 ns sampling time of which 2 ns were discarded for equilibration. Bottom panel: 50 ns sampling time of which 10 ns were discarded for equilibration. (**D**) Autocorrelation times for the interatomic distance between C$_\alpha$ atoms of residues 18 and 45 for simulations in the boxes of different size. The plots in different rows correspond to the interatomic distances used for a harmonic restraint. The shaded areas denote 95% confidence intervals derived from the standard errors over three repeats assuming a normal distribution.

To probe this, we have calculated PMF profiles for dihydrofolate reductase in three boxes (*Figure 5A*) by sampling the M20 loop motion following the description in the original publication (*Babu and Lim, 2020*). In the first example (*Figure 5C* upper panel), we constructed a profile from three umbrella sampling based simulation repeats running 2.5 ns, of which 0.5 ns were discarded for equilibration, for each of the 27 discrete windows along the reaction coordinate (distance between the C$_\alpha$ atoms of residues 18 and 45, as depicted in *Figure 5B*). This rather short sampling time is comparable to the simulations performed by Babu and Lim. Indeed, the free energy profiles obtained from these simulations show an apparent box size dependence. However, in contrast to the original finding, it was the middle box (8 nm edge length) that showed a different behaviour, which appears statistically significant within the limited sampling at hand.

We subsequently extended the simulations 20 times, reaching 50 ns per window along the reaction coordinate. The profiles obtained from these simulations, after discarding the first 10 ns from each window for equilibration, are statistically indistinguishable among the different box sizes (*Figure 5C* bottom panel).

This example once again highlights the caveat of undersampling and its crucial importance for drawing conclusions from the molecular dynamics simulations. To further highlight the manifestation of undersampling, we have calculated autocorrelation functions of the interatomic distance used as a reaction coordinate in every restraint window (*Figure 5D*). For the shorter distances (for brevity, in the figure we show the data for seven windows only), we observe autocorrelations significantly larger than zero pertaining for up to 10 ns. The autocorrelations from the short 2.5 ns simulations follow different trends and are significantly different from those obtained from the longer 50 ns runs. It is evident that analyzing the autocorrelations from the short trajectories may lead to the impression that the 2.5 ns simulation is readily converged, as the local free energy minimum is properly sampled and the trajectory appears to be memoryless. Unfortunately, this situation reveals the nature of the problem being akin to that of Zeno's paradoxes (Aristotle, 350BC, Physics, Book VI): it is only possible to deduce that a simulation is not converged after obtaining a better converged simulation. The longer trajectories reveal that there are other relevant free energy minima, the sampling of which uncovers longer autocorrelation times and subsequently has a pronounced effect on the free energy profiles. The free energy profiles obtained from the trajectories of 10 ns (*Figure 5C* middle panel) already show that the average values for the three box sizes approach one another and the remaining differences are largely within the range of uncertainty.

Furthermore, this case illustrates the importance of generating independent samples and taking into account autocorrelation times to obtain unbiased estimates. For the profiles and associated uncertainties in *Figure 5C* (upper panel), 95% confidence intervals from three independent repeats erroneously indicated statistically signifcant difference between the simulation boxes. This underestimation of the standard error appears due to remaining dependence between the three repeats, as they all started from the same initial structure. The short overall sampling time with a brief equilibration phase were insufficient to remove autocorrelations at longer time-scales. The sufficient equilibration time is system dependent: as we showed before in *Figure 4C*, for alanine dipeptide even shorter simulations were sufficient to obtain reliable error estimates.

## Kinetics: alanine dipeptide

In addition to possible thermodynamic effects, it is also of interest to investigate if the analysis of the kinetics of conformational transitions is affected by the obtained statistics. Here the current literature status is particularly confusing. For the same simulation system, differences in transition times between single repeats have been reported and interpreted (*El Hage et al., 2018*) while at the same time it has been argued that hundreds to thousands of transitions should be sampled for an estimate of kinetic rate constants (*El Hage et al., 2019*).

For the same alanine dipeptide system as used for the thermodynamics investigation described above, we therefore investigated the transition times from the A to the B conformation of the molecule based on a different number of simulation trajectories. Although sampling only ten transitions provides a rather sparse distribution of the transition times (*Figure 6A*), no systematic effect on the rate estimate was observed (*Figure 6B*). Rather, the effect of statistics in this case solely manifests itself in the uncertainty of the rate estimate. As seen in *Figure 6C*, in the case of independent transition trajectories, the uncertainty depends on the number of transitions $N$ as $1/\sqrt{N}$.

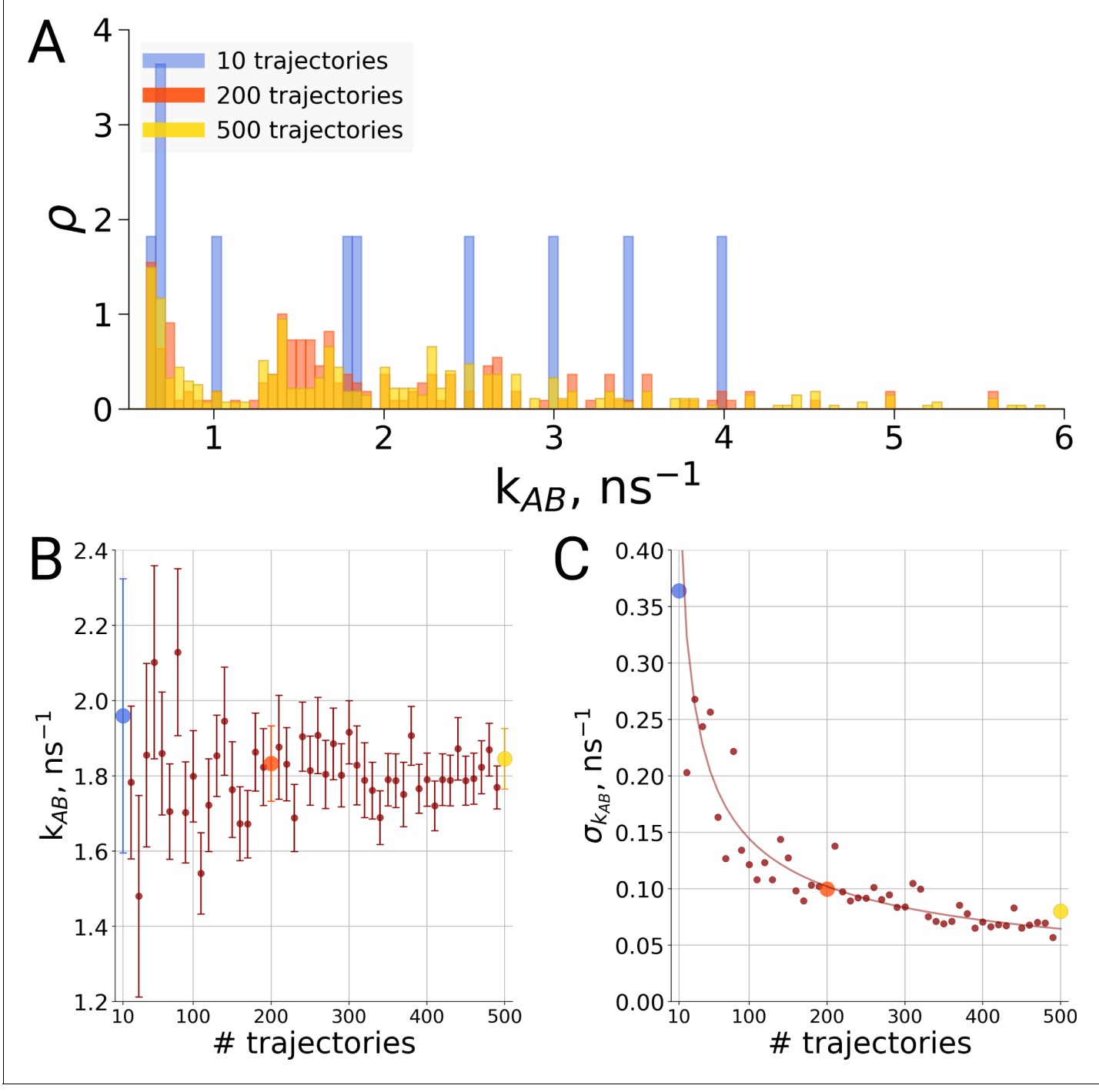

**Figure 6.** Rate estimate dependence on the number of transitions between the two main conformations of alanine dipeptide (backbone dihedral angle $\psi$ is used as a reaction coordinate), the examined conformations are described in detail in *Gapsys and de Groot, 2019*. Panel **A** shows a distribution of transition frequencies for sample sizes of 10, 200 and 500 trajectories. In **B**, the rate estimate is shown as a function of the number of trajectories and in **C** the associated uncertainty is plotted. In **B** and **C** the sample sizes of 10, 200 and 500 are highlighted with blue, orange and yellow spheres, respectively. The line in panel C depicts $1/\sqrt{N}$, scaled to the least squares fit uncertainty.

Thus, to resolve the question of how many transitions are required for a rate estimate, it depends on the required precision for the question at hand. For an order of magnitude estimate, a single transition time may be sufficient, whereas multiple repeats are necessary to obtain a higher precision. Thus, for the general case there is not a single rule of thumb for the required number of

repeats. However, for the frequently occurring case where the question is about rate *differences* when two scenarios are compared to each other (such as a wildtype protein and a point mutation or two simulation box sizes), there is a straghtforward approach, as discussed below.

## Kinetics: hemoglobin

The original discussion on a possible box size dependence in biomolecular simulations was prompted by an investigation into quaternary transitions in human hemoglobin, where in apo simulations of the protein, spontaneous transitions from the T to the R state were observed and an effect of the simulation box size on the transition probability was reported (*El Hage et al., 2018*). Also here it is illustrative to investigate the role of statistics. *Figure 7B* shows the distribution of endstates after one microsecond of simulation for 21 repeats in three different box sizes. For all studied box sizes, some repeats were found to make the transition, defined as ending up closer to the R state, whereas other repeats were found to remain closer to T.

Figure 7C illustrates the effect of limited statistics, and quantifies the probabilities to conclude that only transitions in particular box sizes would take place, if only one repeat per case had been carried out. Although the conclusion that the transition can take place in all three box sizes (consistent with the obervations from all 21 repeats) represents the most probable case also all other scenarios could be picked with a substantial probability, illustrating the risk of working with N = 1 statistics.

## Kinetics of hemoglobin: frequentist inference

Also for the hemoglobin case it is of interest to investigate how the estimated transition time depends on the number of repeats for the different studied box sizes. We present both a frequentist as well as a Bayesian analysis to address the issue. *Figure 8* shows the estimated transition rate (expressed as the estimated time to leave the T state) for the three studied box sizes and for different numbers of repeats. As for the alanine dipeptide case (*Figure 6*), the estimated uncertainty decreases with the number of repeats $N$ as $1/\sqrt{N}$.

In this framework, the question of whether or not the transition time is affected by the simulation box size is addressed as follows. The null hypothesis in this case is that there is no difference between the compared cases and thus the samples are drawn from the same distribution. This null hypothesis can only be rejected if statistically sufficient evidence is provided to demonstrate that the two samples have been drawn from two different distributions. For the data at hand, no significant differences between the studied box sizes are observed for any number of repeats. Hence, the null hypothesis that all samples were drawn from the same distribution cannot be rejected. Accordingly, the hypothesis that the T to R transition time is affected by the simulation box size is not supported by the data.

## Kinetics of hemoglobin: Bayesian inference

The frequentist inference based hypothesis testing, as formulated above, only concludes that there is no sufficient evidence to reject the null hypothesis. It does not, however, allow one to quantify the statistical significance of the opposite, that is, that the kinetics of hemoglobin in differently sized boxes are governed by one process. Such an evaluation can be achieved by following Bayesian inference methodology. Another advantage of the Bayesian approach is that it allows making quantitative statements about the rates when only one or even no transitions are observed throughout a simulation.

In the Bayesian framework, the rate estimates depend on the chosen prior distribution (more details in Materials and methods). Following *Ensign and Pande, 2009*, we chose both a uniform prior as well as Jeffreys prior for the rate estimates. From the generated hemoglobin trajectories, for the simulations in each box, we have extracted the number of transitions from T to R state, as well as the time required to make a transition (or the full trajectory time, in case no transition is made). This information allowed constructing posterior rate constant distributions (see Materials and methods and *Ensign and Pande, 2009*). As shown in *Figure 9*, whereas the choice of prior has a large influence on the probability distribution for the rate for single simulations, this effect is much smaller for 10, 20 and 100 repeats.

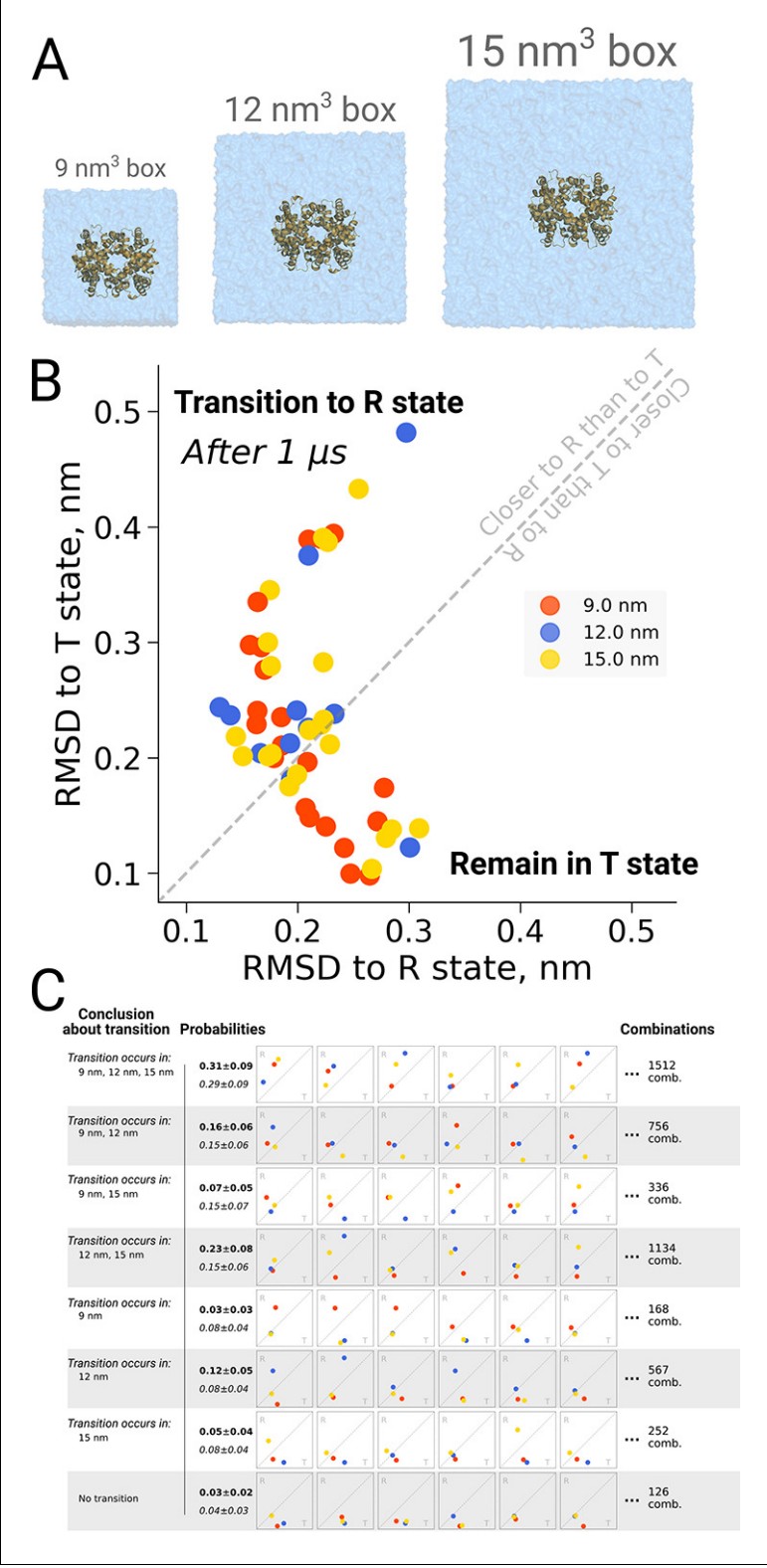

**Figure 7.** Hemoglobin kinetics. The simulations of hemoglobin were performed in 3 different simulation boxes (**A**). Panel **B** depicts the end states in terms of RMSD to the R and T states after 1 μs of simulation. Here, we used the data from *El Hage et al., 2018* and *Gapsys and de Groot, 2019*. In total, 21 simulations were considered in the 9 nm and 15 nm boxes, and 11 simulations in the 12 nm box. In panel **C** we illustrate the numerous combinations that could be obtained from these simulations if only one simulation per box had been

*Figure 7 continued*

performed. Each of those individual combinations could lead to one of 8 conclusions about the box size dependence of hemoglobin transitions: the probabilities for these conclusions based on the currently used data are provided in bold font. We also provide probabilities (italic font) for reaching the conclusions with a condition that the same transition in each of the simulation boxes occurs with the same probability (this probability was calculated by considering the data from all the simulation boxes together).

---

The Bayes formalism provides a framework to estimate if two distributions differ significantly from each other (details in Materials and methods section). As illustrated in *Figure 10*, three scenarios can be distinguished. For a Bayes factor (odds ratio) of around one, no distinction can be made and both samples might have been drawn either from the same or two different distributions. An illustration of this scenario is the case of a single simulation of hemoglobin, where in 9 nm box a transition is observed after 0.466 µs, while for the larger box of 15 nm no transition occurs in 1 µs (*Figure 10* left panel). For Bayes factors in this range between 0.33 and 3 Jeffreys has suggested the label ''barely worth mentioning'' as no distinction can be made between a single or different distributions.

For a Bayes factor of less than 0.33, there is strong evidence that both samples were drawn from a single distribution. This case is illustrated in the middle panel of *Figure 10*, where we compare 100 hemoglobin simulations in 9 nm box with 20 trajectories in 15 nm box.

The third scenario of Bayes factor larger than three would strongly indicate that two different distributions were underlying the samples. None of the hemoglobin simulations, however, have provided evidence to illustrate such a case, therefore, we constructed an artificial synthetic data sample as an illustration (*Figure 10* right panel). To reach the scenario of strong evidence for two disparate processes generating the distributions, we had to artificially replicate three times the observations that in 9 nm box a transition occurs in 0.466 µs and no transition occurs in 15 nm box within 1 µs.

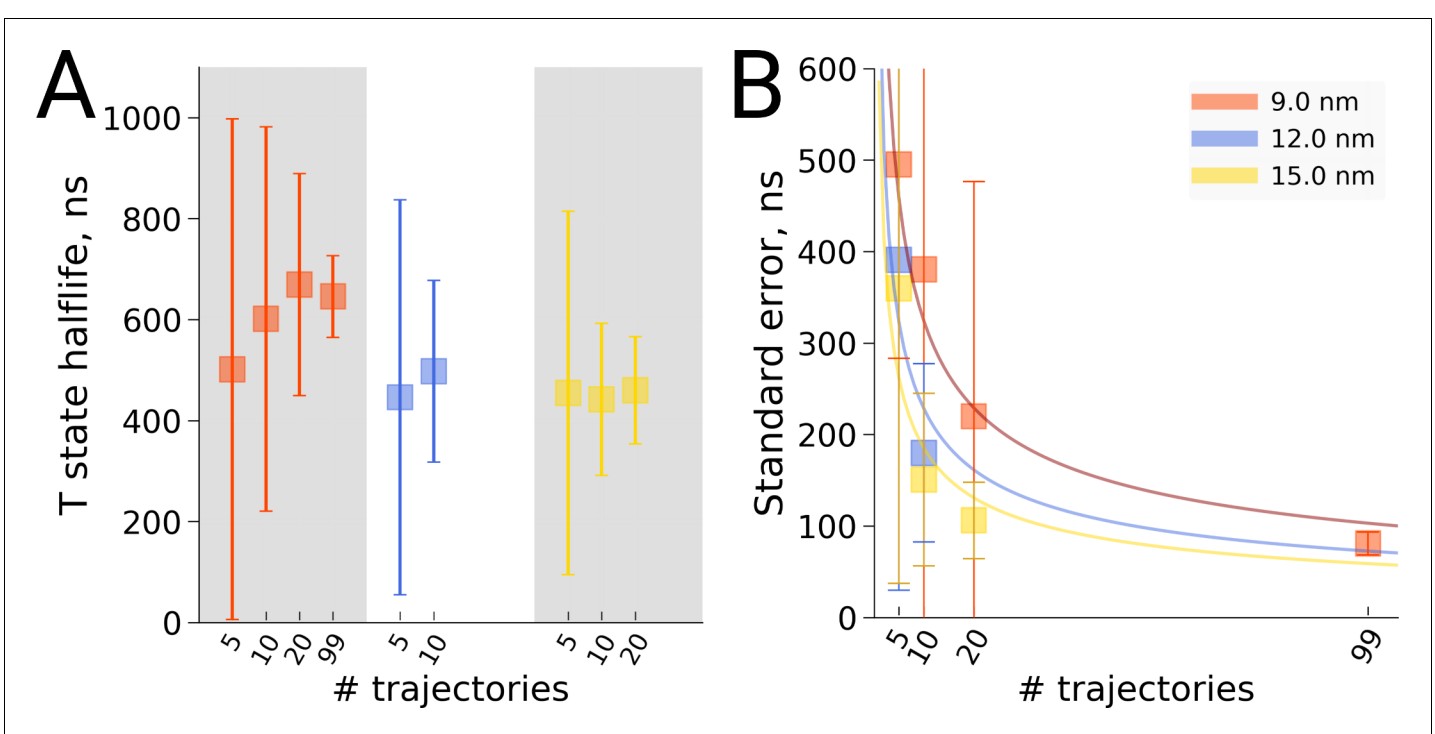

**Figure 8.** Hemoglobin T state half-life statistics. The simulations in 3 differently sized boxes were considered (**A**). The average half-lives together with the associated standard errors were estimated for various numbers of trajectories. In panel B we demonstrate the $1/\sqrt{N}$ behaviour of the standard errors with respect to the number $N$ of the trajectories considered. The error bars in panel **B** denote standard errors indicating that the differences between the standard errors of the estimated half lives are statistically insignificant.

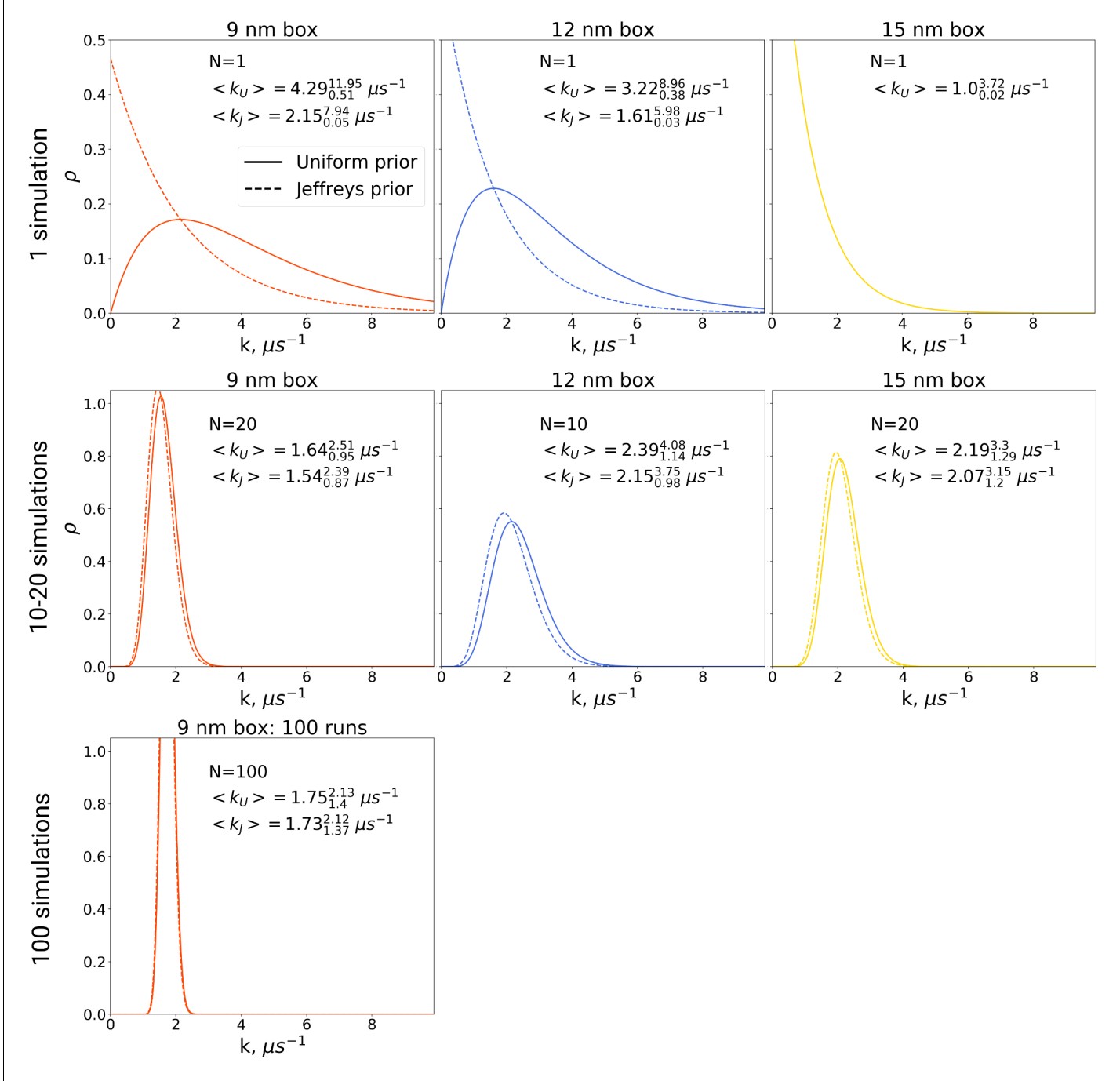

**Figure 9.** The rate constants for hemoglobin's T to R transitions were estimated based on Bayesian formalism following the theory by *Ensign and Pande, 2009*. The figure shows posterior distributions of the rate constants. Two priors were used: uniform (solid line, subscript *U*) and Jeffreys prior (dashed line, subscript *J*). Expected rate estimates with credible 95% intervals are reported in each panel. *N* marks the number of simulations used to construct the respective distributions. With more simulations performed, the rate constant distributions become narrower. As the Jeffreys prior is only applicable for the cases where at least one barrier crossing event is observed, for the *El Hage et al., 2018* hemoglobin simulations in 15 nm box, where no transitions occurred (upper right panel), we only report the rate distribution based on the uniform prior.

Applied to the hemoglobin transition statistics in the different box sizes, this yields the categorization listed in *Table 1*. All comparisons of different box sizes involving single simulation repeats (first three rows of *Table 1*) result in Bayes factors of close to one, meaning that for the associated cases it cannot be distinguished if they stem from a single or two different distributions. This

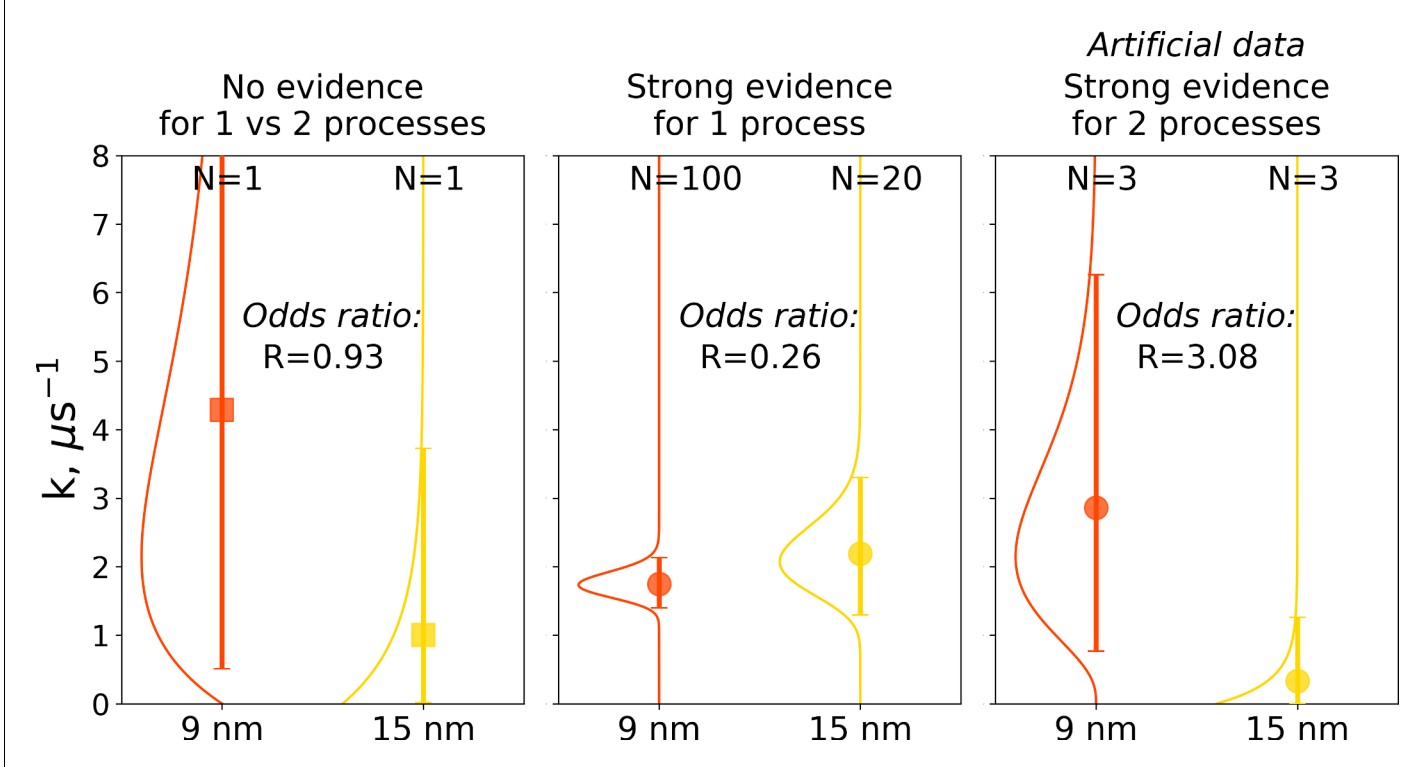

**Figure 10.** Illustration of Bayes factors (odds ratio) for three scenarios. The panels show distributions, together with means and 95% credible intervals for hemoglobin's T to R transition rate constant. The distributions were calculated based on the Bayesian formalism using a uniform prior. In the left panel, data from N = 1 simulations was used: the transition was observed after 0.466 μs in the smaller 9 nm box, but no transition in the larger 15 nm box occurred in 1 μs. In this case, the odds ratio is ~1 indicating that the data provide no evidence to make a conclusion whether the kinetics in two boxes is governed by one or two disparate processes. The middle panel uses data from 100 simulations for the 9 nm box and 20 trajectories in 15 nm box: numerous transitions in both boxes have been observed within 1 μs. In this scenario, the odds ratio is lower than 0.33, hence providing strong evidence supporting the claim that the kinetics in both boxes is governed by one process. While distributions for the left and middle panels came directly from the simulation data reported in *El Hage et al., 2018*, *Gapsys and de Groot, 2019* and from this work, for an illustration in the panel on the right, we needed to resort to an artificial hypothetical case, as none of the simulation data showed evidence for two separate processes governing kinetics in boxes of different size. Therefore, we constructed a synthetic data set where the observations that a transition in 9 nm box occurs in 0.466 μs and no transition happens in 15 nm box within 1 μs were repeated three times each. This resulted in an odds ratio larger than 3, providing strong evidence that the kinetics in the two boxes is governed by two distinct processes.

matches the frequentist outcome above in that the data do not support a conclusion of a box size effect on the T to R transition in hemoglobin. When more statistics are included, the situation changes and smaller Bayes factors are found. Whereas for 9 vs 12 nm and 9 vs 15 nm the outcome remains ambiguous for N = 10,20, for the remaining cases of 9 vs 12 nm, 9 vs 15 nm and 12 vs 15 nm, the suggested interpretation from *Jeffreys, 1961*; *Kass and Raftery, 1995* is that there is substantial evidence that these samples were drawn from a single distribution.

Please note that there is an important difference between these two classes in their implications. Whereas the category with Bayes factors with values between 0.33 and 3 indicate that there is insufficient evidence to justify a conclusion of a box size effect, there might still be such an effect. The three cases with higher simulation repeat numbers (*Table 1*) with a Bayes factor of 0.33 or lower in fact provide substantial evidence that all transition times were drawn from a single distribution, thus providing evidence that the simulation box size does not affect the transition time of the T to R transition in hemoglobin.

## Box size effects on solvent kinetics and thermodynamics

The examples presented so far indicate that the simulation box size effects on both the thermodynamics as well as kinetics of biomolecules are negligible in the studied regime. For the water dynamics, however, there are well documented cases in the literature demonstrating and explaining the

**Table 1.** Odds ratio (Bayes factors) for assessing whether the transitions in simulations of differently sized boxes were governed by disparate processes.

The interpretation of the Bayes factors follows the description by *Jeffreys, 1961*; *Kass and Raftery, 1995*.

| Box size | Number of | Odds favouring | Jeffreys conclusion |
|---|---|---|---|
| | simulations | two processes | |
| 9 nm vs 12 nm | 1 vs 1 | 1.25 | barely worth mentioning |
| 9 nm vs 15 nm | 1 vs 1 | 0.93 | barely worth mentioning |
| 12 nm vs 15 nm | 1 vs 1 | 1.03 | barely worth mentioning |
| 9 nm vs 12 nm | 20 vs 10 | 0.43 | barely worth mentioning |
| 9 nm vs 15 nm | 20 vs 20 | 0.38 | barely worth mentioning |
| 12 nm vs 15 nm | 10 vs 20 | 0.33 | substantial evidence for one process |
| 9 nm vs 12 nm | 100 vs 10 | 0.33 | substantial evidence for one process |
| 9 nm vs 15 nm | 100 vs 20 | 0.26 | substantial evidence for one process |

underlying reasons of the box size effects. A small effect on the diffusion of bulk water is observed due to periodic boundary effects, as described by *Yeh and Hummer, 2004*. Applying a hydrodynamic correction derived by Yeh and Hummer removes any box size effect on the diffusion coefficient in bulk water. For the box sizes equivalent to those used for hemoglobin simulations in this work, the effect on the pure water diffusion is shown in *Figure 11B* (triangles).

It is of interest to investigate how this effect translates in a case of a solvated biomolecule. This analysis is complicated by the fact that the biomolecule restricts the movement of water, an effect that is strongest at the surface of the biomolecule and gradually decreases with increasing distance. In comparing boxes of increasing size, with more and more bulk-like water, the relative number of water molecules restricted by the protein thus changes, limiting the insight that can be gained from a simple averaging over all solvent molecules. This is illustrated in *Figure 11B*, in which the average water diffusion constant is shown for hemoglobin simulation boxes of different size (square symbols). A relatively strong effect is visible, but as laid out above, without further analysis it remains unclear to which extent this is explained by a difference in protein-to-water ratio and to which extent it truly reflects a difference in water diffusion.

To dissect the two effects, in principle the water diffusion constant could be analyzed as a function of the distance to the biomolecular surface. However, water molecules move during the analysis period, changing their distance to the surface and with that their level of restriction, hence challenging the analysis. We therefore chose to use an alternative approach in which we take the largest box of 15 nm as a reference and extrapolate the average diffusion constant of the other two box sizes to the same size of 15 nm by assuming that only bulk water is added (*Figure 11A*). The diffusion constant calculated for the water molecules in the smaller 9 nm and 12 nm boxes was combined by means of a weighted average based on the volume of the added bulk water and the bulk water diffusion coefficient to compute the extrapolated value at 15 nm box size.

As can be seen in *Figure 11B* (circles), the original strong effect from the straight averages (squares) has vanished, indicating that the original apparent effect was largely dominated by a normalization artifact arising from a difference in protein-to-water ratio. It should be noted that the applied extrapolation procedure is approximate, as here we have used simple averaging of non-corrected diffusion coefficients. There is a remaining small trend visible and it will be interesting to analyze in the future if this stems from our crude extrapolation procedure or if it reflects an effect of the periodic boundary conditions.

In addition to water dynamics, also water thermodynamics might in principle be affected by the box size. As with the diffusion constant, a straightforward analysis such as a solvent radial distribution function (RDF) is complicated by normalization issues, which can lead to artifacts when comparing boxes of different size (*El Hage et al., 2018*). The issue arises due to the fact that RDF is normalized by the particle density (in the current case, water density) in the box. In an infinitely diluted system, the density would approach the value of bulk water. In a realistic simulation box,

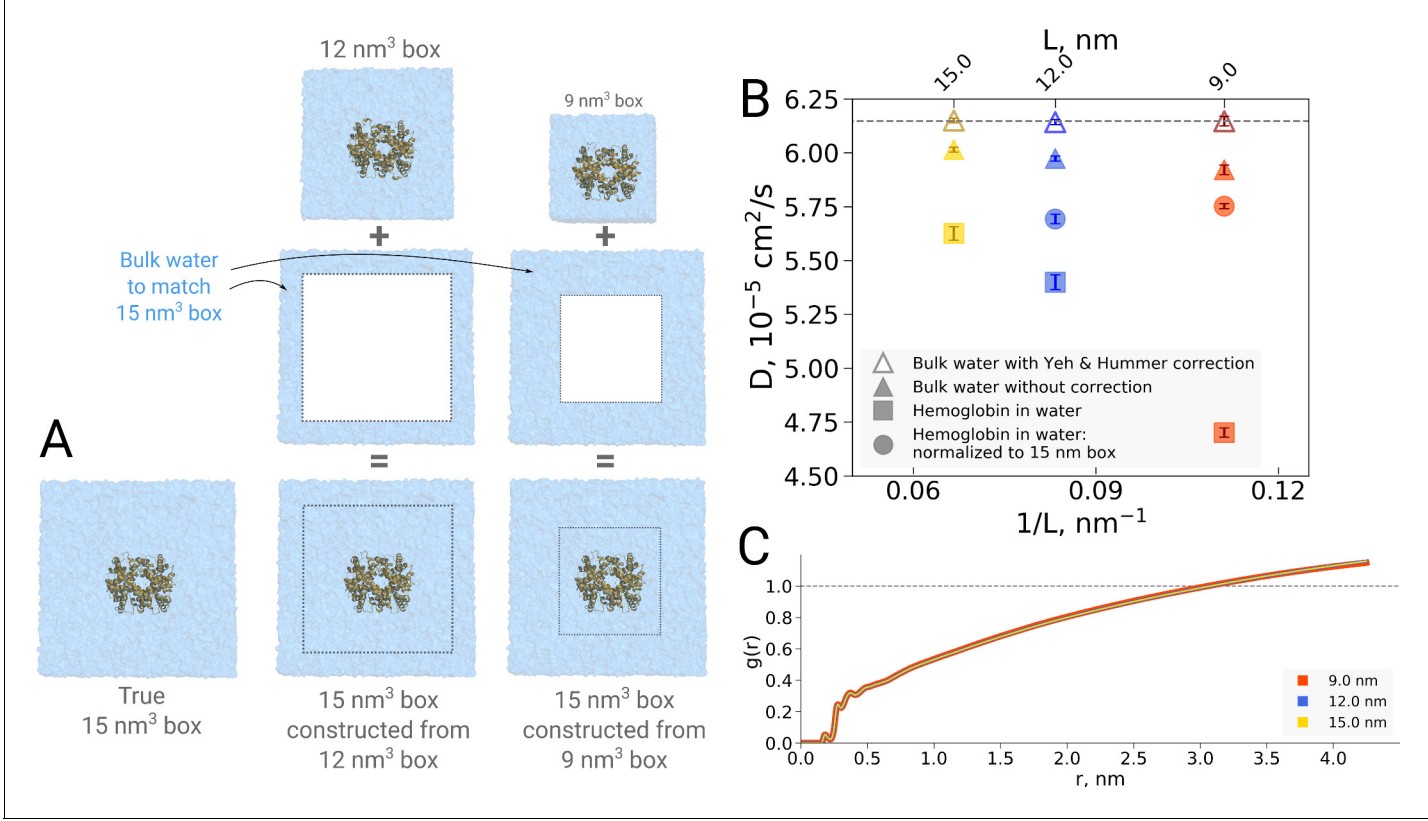

**Figure 11.** Normalization effects in the analysis of differently sized simulation boxes. (**A**) Schematic illustration of the approximate normalization of water diffusion coefficients in smaller boxes against the largest box. Note that no actual rebuilding of simulation boxes was performed, but only the water diffusion coefficients in the smaller boxes were proportionally combined with the bulk water diffusion coefficients. (**B**) Water diffusion in simulation boxes of varying size. A small effect on the diffusion of bulk water (in the pure water simulations) is observed due to the periodic boundary effects, as described by **Yeh and Hummer, 2004** (visualized as triangles). For the Yeh and Hummer correction, a shear water viscosity of $3.08 \times 10^{-4}$ kg $m^{-1}$ $s^{-1}$ was used. A much stronger effect on the water diffusion constant is observed when a large protein, in this case hemoglobin, is added to the simulation box (square symbols). The apparent strong box size effect manifests due to the fact that only little bulk water (capable of diffusing with the bulk-like diffusion constant) is present in the smaller boxes. To account for the difference in protein to water ratio the smaller boxes (9 nm and 12 nm) were renormalized to the level of 15 nm box by employing the value of bulk water diffusion weighted by the difference in the number of water molecules between the smaller boxes and 15 nm box (circle symbols). The dashed line markes the bulk water diffusion value for water simulation in 9 nm box. (**B**) Water radial distribution function (RDF) for differently sized hemoglobin boxes, normalized to a sphere of 8.5 nm diameter.

however, a fraction of its volume that is occupied by the protein is not accessible to water. This way, the calculated RDF is larger than the true RDF by $V_{box}/(V_{box}-V_{protein})$. This ratio depends on the box size, in turn, making the normalization of RDF box size-dependent.

To overcome this issue, we have previously used a procedure of cutting out boxes of constant size from differently sized simulation boxes, rendering the RDF analyszd from the cut-out boxes comparable. This analysis showed that the strong apparent box size effect on the solvent RDF initially derived from the different box sizes (**El Hage et al., 2018**) vanished upon consistent normalization (**Gapsys and de Groot, 2019**).

However, as the procedure of cutting out subsystems from a simulation trajectory is a somewhat crude and ad hoc procedure, we now implemented a spherical RDF normalization that does not rely on subsystem extraction from larger boxes. As can be seen in **Figure 11C**, the result of this radial normalization is consistent with our previous observation (**Gapsys and de Groot, 2019**) as well as the anthracene solvation presented in **Figure 1** and it confirms that no significant box size effects are observed in terms of protein solvation and, by implication, the hydrophobic effect.

The combined results from *Figure 11* demonstrate the established (*Yeh and Hummer, 2004*) mild box size dependence of bulk water dynamics. In contrast, no evidence for a thermodynamic effect on the solvent shell was found.

## Discussion

In the work we have not observed any statistically significant box size effects on the kinetics and thermodynamics of biomolecules. It should be noted, however, that all box sizes studied in this work included a solvent shell of at least 1 nm in size, or at least three solvation layers between the studied biomolecule and the box edge. Beyond that range only minimal box size effects should be expected, as shown, but naturally box size artifacts may occur for smaller boxes due to (indirect) self-interaction caused by the periodic nature of typical molecular dynamics boxes (*van Gunsteren and Berendsen, 1990*; *Mehra and Kepp, 2019*). Therefore the box-size independence demonstrated in this work is limited to box sizes that exceed a distance of 1 nm between the biomolecule and the box edge.

Similarly, simulations in particularly small boxes may have a profound effect on the lateral lipid diffusion, resulting in significant correlations in lipid motions (*Klauda et al., 2006*). Furthermore, lateral diffusion of lipids, as well as membrane proteins embedded in a lipid bilayer depend on the simulation box size and box asymmetry due to the hydrodynamic effects (*Vögele et al., 2018*), Hydrodynamic effects also induce the rotational diffusion dependence on the simulation box size (*Linke et al., 2018*). Analytical hydrodynamic corrections for the latter effects have been proposed (*Vögele and Hummer, 2016*; *Linke et al., 2018*) in a similar fashion to the correction for the bulk water diffusion by *Yeh and Hummer, 2004*.

Among other box size related caveats that need to be considered when performing simulations in a periodic system where electrostatics is treated by means of Ewald summation, is the overall charge of the system. For a charged simulation box, a uniformly distributed neutralizing background charge will be introduced automatically by the Ewald methods, thus the charges in the simulated system will experience a potential which will depend on the solvent volume (*Lin et al., 2014*) and dielectric constants of the solvents (*Hub et al., 2014*). While posthoc corrections to the free energies calculated in the charged periodic systems exist (*Rocklin et al., 2013*; *Hub et al., 2014*; *Lin et al., 2014*; *Reif and Oostenbrink, 2014*), a generalize advice to obtaining reliable dynamic trajectories would be, if possible, to neutralize the simulation system.

All in all, in the work we aimed at collecting a comprehensive set of examples illustrating that a reliable uncertainty estimation is not a mere nuisance, but an essential part of good scientific practice. Ignoring to obtain error estimates and relying on single realizations of processes stochastic in nature may lead to drawing incorrect conclusions, as for example showcased in *Figure 1*. A previous work by *El Hage et al., 2018* serves as clear illustration of this when interpreting a set of single (no replicas) simulation trajectories. Upon generation of a larger data sample, the previous conclusions of El Hage et al. are shown not to hold (*Figure 7*). It is our hope that this work will provide a reference for the scientific community to identify cases that do not meet scientific standards (*Pezzella et al., 2020*).

## Conclusion

In this work we have systematically investigated the effect of the usage of statistics on the interpretation of results from molecular simulation. Due to the stochastic nature of most commonly used sampling integrators, it is of critical importance that sampling convergence is demonstrated. This can be achieved in single runs if the slowest degree of freedom relevant for the property of interest is reversibly sampled multiple times. Alternatively, multiple repeats can be carried out to check for reproducibility and convergence. The use of multiple repeats has many additional advantages such as a straightforward error estimate obtained from the spread among individual repeats. This is frequently a valid approach, provided that the individual repeats can be considered statistically independent. A practical advantage of multiple repeats is its trivial parallelization. In the absence of multiple repeats, there is a substantial risk of lack of reproducibility due to the fact that the results from individual runs may have happened by chance and are not representative of a larger set. Such anecdotal evidence may be useful for hypothesis generation but should be followed up by a more thorough statistical approach. The examples provided in this work illustrate that an initial indication

of an apparent box size effect in molecular dynamics simulations based on single repeats was not confirmed when repeated with an order of magnitude more statistics.

## Materials and methods

### Anthracene solvation free energy calculations

An automated procedure was used to assign the CGenFF based topology for anthracene (*Vanommeslaeghe and MacKerell, 2012*; *Vanommeslaeghe et al., 2012*). The molecule was placed in cubic boxes with the following edge length: 0.5, 0.6, 0.8, 1.0, 1.5 and 2.0 nm. The molecule was solvated with TIP3P water (*Jorgensen et al., 1983*).

To calculate hydration $\Delta G$ we used non-equilibrium free energy calculation setup (*Gapsys et al., 2015a*). Firstly, we performed equilibrium simulations of anthracene coupled to water via (*van Gunsteren and Berendsen, 1990*) van der Waals and electrostatic interactions, (*Lindorff-Larsen et al., 2012*) van der Waals interactions only and (*Knapp et al., 2018*) in decoupled state. Each equilibrium simulation was of 6 ns. Afterwards, 80 frames were extracted equidistantly in time from each generated trajectory after discarding first 2.3 ns. Fast non-equilibrium transitions of 0.2 ns each were started from the extracted simulation snapshots. The transitions were performed from state *van Gunsteren and Berendsen, 1990* to *Lindorff-Larsen et al., 2012* and from *Lindorff-Larsen et al., 2012* to *van Gunsteren and Berendsen, 1990* to calculate $\Delta G_Q$; the transitions from state *Lindorff-Larsen et al., 2012* to *Knapp et al., 2018* and from *Knapp et al., 2018* to *Lindorff-Larsen et al., 2012* allowed obtaining $\Delta G_{vdw}$. The solvation free energy reported in *Figure 1B* was calculated as the sum of two contributions. The free energy differences were calculated from the work values of non-equilibrium transitions relying on the Crooks fluctuation theorem (*Crooks, 1999*) based on the maximum likelihood estimator (*Shirts et al., 2003*) as implemented in pmx (*Gapsys et al., 2015b*). The whole procedure was repeated 20 times for every box allowing to estimate standard error from the standard deviation of independent repeats. The standard errors for the $\Delta G$ for individual repeats (*Figure 1B* right panel) were calculated by means of bootstrap.

The simulations were performed with Gromacs (*Abraham et al., 2015*) version 2018. The stochastic dynamics integrator was used to integrate the equations of motion at 298 K with a friction of constant of 0.5 ps$^{-}$1. The pressure was kept at 1 bar by means of a Parrinello-Rahman barostat (*Parrinello and Rahman, 1981*) with a time constant of 5 ps. All anthracene bonds were constraint by means of the LINCS algorithm (*Hess et al., 1997*) of the order 6. Particle Mesh Ewald (*Darden et al., 1993*; *Essmann et al., 1995*) was used for long range electrostatics with a direct space cutoff at 1.0 nm and a Fourier spacing of 0.12 nm. The van der Waals interactions were switched off between 0.9 and 1.0 nm. Dispersion correction for energy and pressure was applied.

### $G_B$ protein solvation free energy calculations

Solvation free energies of the $G_B$ protein were computed with the Gromacs version 2018. All the simulations were carried out in the Charmm36m force field (*Huang et al., 2017*) with TIP3P water (*Jorgensen et al., 1983*). The starting structure for simulations was extracted from the NMR ensemble 2LHD (*He et al., 2012*) (model 3). The structure was solvated and energy minimized. For all the subsequent $G_B$ simulations, every protein atom was kept restrained with the restraint force constant of 1000 kJ/mol/nm$^2$. Free energy calculations were initialized from an equilibrated set of conformations. Equilibration was performed in the largest simulation box by running Hamiltonian replica exchange simulations of the uncharged protein, performing van der Waals interaction annihilation only. The end states of these simulations were used to start further production runs. If required, smaller simulation boxes were constructed from the largest box by discarding waters distant from the protein. For the free energy calculations, charge and van der Waals annihilation was performed by explicitly adjusting protein's topology. The stratification protocol along the alchemical $\lambda$ coordinate used 23 discrete windows for the charge annihilation and 32 windows for the annihilation of van der Waals interactions. Every window used 10 ns sampling, of which the first 2.5 ns were discarded from analysis. For the annihilation calculations in water we performed five independent simulation repeats and an additional simulation with the Hamiltonian replica exchange. Calculations in vacuum were repeated three times. The final free energies were estimated by the multistate Bennet

acceptance ratio (MBAR) (*Shirts and Chodera, 2008*) estimator as implemented in the *alchemical_analysis* (*Klimovich et al., 2015*).

The $G_B$ protein solvation calculations used a similar simulation setup to the anthracene simulations, except for a longer electrostatic direct space cutoff of 1.3 nm. Also, van der Waals interactions were switched off between 1.2 and 1.3 nm. Only hydrogen containing bonds were constrained by means of LINCS.

### Dihydrofolate reductase

For the dihydrofolate reductase simulation setup, we closely followed the protocol described by *Babu and Lim, 2020*. Once solvated and neutralized, the systems were first equilibrated for 1 ns with position restraints on heavy protein atoms, followed by 1 ns unrestrained equilibration. The free energy profiles were constructed from equilibrium simulations at 27 discrete windows (umbrella sampling) along the reaction coordinate defined as a distance between the $C_\alpha$ atoms of residues 18 and 45. A harmonic restraint with a force constant of 4184 kJ/mol/nm$^2$ was applied to the drive distance from 0.4 to 1.7 nm by a 0.05 nm increment. Three independent simulation repeats were performed for each of the three simulation boxes reaching 50 ns per window. The final PMF profile was calculated by means of MBAR estimator (*Shirts and Chodera, 2008*).

Here again we used a similar set of simulation parameters to the anthracene case, except for integrating equations of motion with the leap-frog integrator. The temperature was kept at 300 K by means of the velocity rescaling thermostat (*Bussi et al., 2007*) with the time constant of 0.2 ps. The electrostatic potential was shifted to reach zero at the cutoff value of 1.2 nm. The van der Waals interactions were modified by smoothly switching the forces to zero between 1.0 and 1.2 nm. Hydrogen containing bonds were constrained with LINCS.

### Alanine dipeptide simulations

The alanine dipeptide data were taken from previously published trajectories (*Gapsys and de Groot, 2019*). To summarize briefly, we have prepared four simulation setups in cubic boxes with an edge of 3.0, 5.0, 7.0 and 9.0 nm. For each of the boxes we performed 10 independent simulations of 1 μs each. More details on the simulation parameters can be found in *Gapsys and de Groot, 2019*.

The free energy profiles were calculated for every trajectory and aligned by setting the minimal value along Ψ dihedral angle to ΔG = 0 kJ/mol. The standard errors were calculated from the standard deviations of independent simulations.

For the transition rate calculations, the trajectories were divided in 6000 sub-trajectories of 1.6 ns each. The trajectory length was chosen such that on average, one transition takes place per sample, comparable to the hemoglobin simulations of 1 microsecond each. Samples of sizes 10, 200 and 500, as well as regular intervals in between, were selected randomly from the generated trajectories. Transition times were recorded as described previously (*Gapsys and de Groot, 2019*) and the mean and standard error for each sample size was computed.

### Hemoglobin simulations

The hemoglobin simulations were taken from previous work (*Gapsys and de Groot, 2019*). In the current work we analyzed simulation trajectories generated in the cubic boxes with an edge of 9, 12 and 15 nm based on the simulation protocol by *El Hage et al., 2018*. For the boxes with an edge of 9 nm and 15 nm, 20 trajectories have been simulated, while for the box with an edge of 12 nm, 10 trajectories were used. In the current work 80 additional repeats of simulations in the 9 nm box were added using *El Hage et al., 2018* simulation parameters and setup to yield a total of 100 simulations for this box. The duration of each simulation for every box size was 1 μs.

### Frequentist rate estimation

For hemoglobin simulations in each box size, we divided the overall pool of generated trajectories into smaller sub-samples, e.g. for the 9 nm box we randomly created sets of size 5, 10, 20 and 99. Within each of these sets we performed 1000 bootstrap runs, where for each run a survival curve was calculated (*Gapsys and de Groot, 2019*). Exponential fitting to a survival curve yields the time constant, which can be related to the half-life by $t_{1/2} = \tau \ln 2$. Standard error was calculated as a

standard deviation of a bootstrapped distribution. The described approach, however, does not allow estimating the half-life for those cases, where no transition is observed: for such situations, Bayesian method for rate estimation is to be used.

## Bayesian rate estimation

The Bayesian rate estimation used in this work was based on the theoretical framework developed by *Ensign and Pande, 2009* for single exponential kinetics. Here, we provide only a brief summary of the main concepts and equations that were used in the current work. In all the equations, we follow closely the notation by Ensign and Pande.

Bayesian relation is used to express the posterior probability density $p(k|D,I)$ of the rate $k$ as a product of likelihood $l(k|D,I)$ and prior rate distribution $p(k|I)$ ($D$ is the observed transition data and $I$ is other background information about the model):

$$p(k|D,I) \propto l(k|D,I)p(k|I) \tag{1}$$

Subsequently, for $N$ trajectories generated starting from a configuration $X$, the likelihood function is expressed as:

$$l(k|D,X,I) = k^n \mathrm{e}^{-k\Theta} \tag{2}$$

Here $\Theta$ defines the total trajectory time, considering that $n$ trajectories of $N$ make a transition:

$$\Theta = \sum_{i=1}^{n} t_{fi} + \sum_{i=1}^{N-n} t_i \tag{3}$$

The time for a trajectory to make a transition is denoted as $t_{fi}$, while the length of the trajectories not showing a transition is $t_i$. Ensign and Pande derive posteriors for two prior distributions - uniform and Jeffreys. The posterior distribution for the uniform prior is expressed as:

$$p_U(k|D,X,I) = \frac{\Theta^{n+1}}{n!} k^n e^{-k\Theta} \tag{4}$$

The expectation value for this posterior distribution is $\langle k \rangle_U = (n+1)/\Theta$ and the variance is $\mathrm{var}(k)_U = (n+1)/\Theta^2$. The posterior for the Jeffreys prior is:

$$p_J(k|D,X,I) = \frac{\Theta^n}{(n-1)!} k^{n-1} e^{-k\Theta} \tag{5}$$

The expectation value and variance for this posterior are $\langle k \rangle_J = n/\Theta$ and $\mathrm{var}(k)_J = n/\Theta^2$, respectively.

The expressions above illustrate that the posterior distribution based on uniform prior allows inferring rates even for the cases where no transitions have been observed. Jeffreys prior, although being invariant to scaling of $\Theta$, does not provide a defined posterior distribution in the absence of observed transitions. Therefore, using the uniform prior has an advantage that it allows making inferences about rates even when no transitions are observed. In practice, whenever possible, it is recommended to use both priors and compare the generated posteriors: given sufficient evidence by the data, both posteriors converge to the same distribution.

To compare the odds ratio of the two models we, firstly, define two hypotheses: $H_A$: one process with a rate $k$ has generated the data observed in differently sized boxes. $H_B$: the data from different boxes were generated by distinct processes with disparate rates. The expression for Bayes factor (odds ratio) is:

$$\frac{P(D|H_B,I)}{P(D|H_A,I)} = \frac{2}{\pi} \left( \frac{n/\Theta}{n_1^2/\Theta_1^2 + n_2^2/\Theta_2^2} \right) \left( \frac{n_1!n_2!}{n!} \right) \left( \frac{\Theta^{n+1}}{\Theta_1^{n_1+1}\Theta_2^{n_2+1}} \right) \tag{6}$$

### Water diffusion

For the water diffusion calculations, we ensured that the calculated values did not suffer from the heuristic particle position unwrapping scheme for periodic boundary conditions implemented in Gromacs, as described by *von Bülow et al., 2020*. We calculated the diffusion coefficients by dividing

trajectories into smaller chunks, but observed no significant difference. This is likely due to the fact that the approximate critical time for the unwrapping issue to manifest for the hemoglobin simulations is longer than the overall time of our simulations.

### Radial distribution function

We recomputed the solvent radial distribution function for the simulations of different box sizes published previously (*Gapsys and de Groot, 2019*; the simulations followed *El Hage et al., 2018* setup). In the current implementation we added the functionality to use a specified radius for normalization, rather than the default simulation box size. For the presented data in *Figure 11* we used a normalization distance of 4.25 nm radius. The code for this implementation is available under https://github.com/blauc/gromacs/tree/rdf.

## Acknowledgements

This work was supported by BioExcel CoE (www.bioexcel.eu), a project funded by the European Union contracts H2020-INFRAEDI-02-2018-823830 and H2020-EINFRA-2015-1-675728. We thank Christian Blau for the custom RDF implementation and Kresten Lindorff-Larsen for bringing to our attention the Bayesian rate estimation methodology.

## Additional information

### Funding

| Funder | Grant reference number | Author |
| --- | --- | --- |
| European Commission | H2020-EINFRA-2015-1-675728 | Vytautas Gapsys Bert L de Groot |
| European Commission | H2020-INFRAEDI-02-2018-823830 | Vytautas Gapsys Bert L de Groot |
| Max-Planck-Gesellschaft | | Bert L de Groot |

The funders had no role in study design, data collection and interpretation, or the decision to submit the work for publication.

### Author contributions

Vytautas Gapsys, Conceptualization, Data curation, Software, Formal analysis, Validation, Investigation, Visualization, Methodology, Writing - original draft, Project administration; Bert L de Groot, Conceptualization, Formal analysis, Supervision, Funding acquisition, Validation, Investigation, Methodology, Writing - original draft

### Author ORCIDs

Vytautas Gapsys https://orcid.org/0000-0002-6761-7780
Bert L de Groot https://orcid.org/0000-0003-3570-3534

### Decision letter and Author response

Decision letter https://doi.org/10.7554/eLife.57589.sa1
Author response https://doi.org/10.7554/eLife.57589.sa2

## Additional files

### Supplementary files

• Transparent reporting form

### Data availability

Input files and data for all the figures is provided: https://doi.org/10.5281/zenodo.3959198.

The following dataset was generated:

| Author(s) | Year | Dataset title | Dataset URL | Database and Identifier |
|---|---|---|---|---|
| Gapsys V, de Groot B | 2020 | Input files and data for the publication | https://doi.org/10.5281/zenodo.3959198 | Zenodo, 10.5281/zenodo.3959198 |

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
