## [Decision Letter]

**Acceptance summary:**

The value of this manuscript resides in its rigorous analysis and clear presentation of non-trivial methodological issues that are of relevance to all molecular simulation studies. It is anticipated that this work will serve a reference for the growing community of scientists using these types of computational methods.

**Decision letter after peer review:**

Thank you for submitting your article "On the importance of statistics in molecular simulations: thermodynamics, kinetics and simulation box size" for consideration by *eLife*. Your article has been reviewed by three peer reviewers, and the evaluation has been overseen by José Faraldo-Gómez as the Senior Editor. The following individual involved in review of your submission has agreed to reveal their identity: Alan Grossfield (Reviewer #3).

The reviewers have discussed the reviews with one another and the Senior Editor has drafted this decision to help you prepare a revised submission. In the interest of full transparency, the Senior Editor has decided to provide you with the reviewers' reports as received. As you will see, the reviewers have judged that your manuscript is of interest and potentially suitable for *eLife*. However, they also conclude that additional data and revisions to the text will be required before the manuscript can be accepted for publication.

Given this decision, I would like to draw your attention to the changes in the *eLife* revision policy that were made in response to the COVID-19 crisis (https://elifesciences.org/articles/57162). First, because many researchers have temporarily lost access to the labs, we will give authors as much time as they need to submit revised manuscripts. We are also offering, if you choose, to post the manuscript to bioRxiv (if it is not already there) along with this decision letter and a formal designation that the manuscript is "in revision at *eLife*". Please let us know if you would like to pursue this option.

Reviewer #1:

The work by Gapsys and de Groot addresses the critical issues of adequate sampling and proper statistical analysis in molecular simulations, which are essential to provide a rigorous assessment and interpretation of the simulation results. Molecular simulations are necessarily performed on limited-size systems confined in a simulation box, typically under periodic boundaries and it is legitimate to investigate the implications of such finite-size approximation in conventional simulation setups.

In this work the authors analyze the effect of the simulation box size in different types of simulations and bimolecular systems, ranging from small test systems to more realistic biological molecules (i.e. hemoglobin), exemplifying in each case the possible shortcomings arising from sampling and statistical inaccuracies. In the case of hemoglobin, the authors propose rigorous statistical analysis to evaluate the trend of the transition times from T to R protein conformations as a function of the box size for different simulations repeats. This work report an extensive set of examples, is well written and appear very well done from the technical standpoint. Nonetheless in my opinion the manuscript requires a few revisions to improve the clarity of the presentation and to provide a better and more comprehensive fit with the current literature.

One of the main conclusions of this work is that beyond a reasonable threshold the box size does not affect the simulation outcome. While this has been clearly proven for the systems and observables investigated in this study, to provide a broader perspective it is important to mention clear-cut examples in which the effect of the finite box size must be taken into consideration to provide meaningful results and in some case could also lead to artifacts.

For the case of the alchemical calculations reported in the manuscript the authors should mention and discuss documented cases of inhomogeneous systems (e.g. when charges are generated) in which the periodic boundaries could lead to shifts in the calculated free energy unless a proper setup is utilized (Lin et al., 2014). One example of artifact that could be induced by the periodic boundaries is also reported in a previous study by one of the authors of this work (Hub et al., 2014). Another example is the significant system size dependence of the lateral diffusion of lipids (Klauda et al., 2006). Lastly, even for homogenous systems a box size dependence of the rotational diffusion of biomolecules has been recently reported (Linke et al., 2018).

A general question that I have pertains to the examples of Figure 1C and Figure 5C, in which the authors illustrate the anecdotal trends that could be obtained with limited statistics. It could be instructive in these cases to report additional controls that could be taken for assessing lack of statistics. In Figure 1C for example block analysis within each single repeat could already indicate that the reported trends as a function of the box size are not significant. In the case of Figure 5C autocorrelation analysis and/or block analysis could be used to assess the presence of correlations between the repeats (i.e. each single repeat is not equilibrated). Another possibility is to compare the results with a Markov Model analysis, such as TRAM (Wu et al., PNAS June 7, 2016 113 (23) E3221-E3230) or DHAM (Stelzl et al., J Chem Theory Comput. 2017 Dec 12;13(12):6328-6342) that should better handle non equilibrated samples.

I report below more specific clarifications/improvements that need to be addressed.

In the application to the Gb protein the authors compare their results to the work of Asthagiri and Tomar. Here the explicit comparison must be clarified; Asthagiri and Tomar also report a thermodynamic integration calculation to estimate the charging free energy of Gb which is (consistently with this work) box size independent. The box size dependence emerges when using the quasi chemical theory (e.g. to analyze the specific components). Additionally, the charging free energy reported by Asthagiri and Tomar doesn't seem consistent with the values reported in this work.

The Discussion section is more a result section about water diffusion, my suggestion is to move this section in the results and provide an actual discussion, e.g. expanding the Conclusions by including the considerations reported above.

Reviewer #2:

General assessment:

This paper aims to present a methodical demonstration of the importance of obtaining complete sampling statistics from MD simulations for the purpose of estimating thermodynamic or kinetic quantities. As a test case, the authors apply this demonstration to the hypothesis that solvent periodic box size affects the simulation-derived thermodynamics and kinetics of biomolecules. This work is statistically rigorous and provides a clear comparison between results obtained with insufficient sampling and those derived from more extensive sampling. This paper can serve as a good reference for future scientists employing MD simulations who want to make sure their sampling is adequate based on proper statistical methods.

Major comments:

1) The authors show that increased simulation time per window (50 ns instead of 2.5 ns) along the dihydrofolate reductase M20 loop motion reaction coordinate results in indistinguishable PMF profiles by box size. The authors attribute the discrepancies observed when using 2.5-ns sampling times to insufficient equilibration away from the initial structure, as 2.5 ns is not sufficiently greater than the autocorrelation time of the loop motion. While this is a very plausible explanation, the conclusion would be strengthened by explicit calculation of the reaction coordinate autocorrelation time. This is particularly important to include as an example to other scientists who would like to estimate how long their independent simulations need to be to avoid biased estimates that are caused by using the same initial structure.

2) For the kinetics analysis it would be helpful to include more information, including equations or a reference to the appropriate equations, to show how the rate constants and uncertainty were calculated in detail using the frequentist approach. In particular, it is not clear how the trials that did not exhibit any transitions for hemoglobin were accounted for in the calculation of the T state halflife.

Reviewer #3:

I think this is a timely, important, and well-executed study that will be a valuable resource to the community. Insufficient sampling has been the bane of bimolecular simulation since its inception 40 years ago, and this paper is largely framed as a rebuke to a recent, high-profile example.

My biggest complaint with the paper as it stands is not with the work, which is excellent, but with the lack of details in the presentation. As I see it, the real value in this manuscript isn't beating up on a bad paper, the authors already accomplished that, but in teaching the community how to do things better. The paper covers a truly enormous scope, arguably, 3-4 paper's worth, and as a consequence doesn't include all of the details needed to understand their arguments. Most of the information is available in the references, or in the code they distribute (another excellent choice on their part), but to really maximize the didactic value of the paper I think more of the explanation has to be in the main body of the paper. For example:

1) They never said what enhanced sampling method was used in the DHFR calculations, or what the collective variable was. It sounds like umbrella sampling, but they weren't explicit.

2) Figures 8-10 on the kinetics of hemoglobin. There's no real explanation of what they did to generate any of these figures. The error bars in Figure 8 aren't defined, nor is how they were calculated. I'm assuming they did bootstrapping, with different sample sizes, but it should be stated explicitly. It's interesting that the standard errors in Figure 8B are larger for the smaller box, but the authors don't comment on why.

3) Figure 9: there are 2 curves in each panel, presumably one for each prior, but which is which is not stated. In the top right panel, there's only 1 curve, with no explanation. Given that the paper is mostly intended as a teaching tool, I think a brief summary of the Ensign and Pande theory would be helpful, as would an explanation for why you would use to two different priors.

4) Figure 10: You have to read pretty far into the caption before you realize the data which shows 2 distributions is artificial, to prove a point. No details are given to explain how this was done, in the caption or the body of the paper. The authors should either leave it out or provide those details.

5) Very few details are given about the alanine dipeptide calculations.

6) When discussing the protein solvation calculations, the final conclusion (that the appropriate approach to eliminate box-size dependence in vacuum is to use no cutoff at all) is very reasonable, but the data should be shown.

Despite my somewhat lengthy list of complaints, I want to re-emphasize that I think this is an outstanding paper.

---

## [Author Response]

Reviewer #1:[…] One of the main conclusions of this work is that beyond a reasonable threshold the box size does not affect the simulation outcome. While this has been clearly proven for the systems and observables investigated in this study, to provide a broader perspective it is important to mention clear-cut examples in which the effect of the finite box size must be taken into consideration to provide meaningful results and in some case could also lead to artifacts.For the case of the alchemical calculations reported in the manuscript the authors should mention and discuss documented cases of inhomogeneous systems (e.g. when charges are generated) in which the periodic boundaries could lead to shifts in the calculated free energy unless a proper setup is utilized (Lin et al., 2014). One example of artifact that could be induced by the periodic boundaries is also reported in a previous study by one of the authors of this work (Hub et al., 2014). Another example is the significant system size dependence of the lateral diffusion of lipids (Klauda et al., 2006). Lastly, even for homogenous systems a box size dependence of the rotational diffusion of biomolecules has been recently reported (Linke et al., 2018).

We have extended the Discussion section incorporating the description of the cases that are box size sensitive.

“Similarly, simulations in particularly small boxes may have a profound effect on the lateral lipid diffusion, resulting in significant correlations in lipid motions (Klauda et al., 2006). […] While posthoc corrections to the free energies calculated in the charged periodic systems exist (Rocklin et al., 2013,Hub et al., 2014,Lin et al., 2014,Reif et al., 2014), a generalize advice to obtaining reliable dynamic trajectories would be, if possible, to neutralize the simulation system.”

A general question that I have pertains to the examples of Figure 1C and Figure 5C, in which the authors illustrate the anecdotal trends that could be obtained with limited statistics. It could be instructive in these cases to report additional controls that could be taken for assessing lack of statistics. In Figure 1C for example block analysis within each single repeat could already indicate that the reported trends as a function of the box size are not significant.

For every individual data point in Figure 1C (i.e. every free energy estimate) we have calculated a bootstrap based error estimate (visualized in Figure 1D). The large error bars depicting 95% confidence intervals illustrate the danger of interpreting the trends without considering the underlying uncertainties. When the uncertainties are taken into account, it appears that the confidence intervals cover a large range of G values making the judgement about only increasing/decreasing trend statistically insignificant.

“Naturally, if we were to rely on single realizations of molecular dynamics trajectories, it is possible to obtain any type of trend suggesting a box size dependence: in the middle panel of Figure 1B we highlight an arbitrary selection of an upward and downward trends. However, neither trend is statistically significant and merely illustrates the erroneous conclusion that may be drawn from anecdotal evidence. This analysis also clearly illustrates the importance of reporting uncertainty estimates for the calculated observables: depicting confidence intervals for the G estimates (Figure 1B right panel) would help avoiding making ungrounded claims about the depicted trends.”

In the case of Figure 5C autocorrelation analysis and/or block analysis could be used to assess the presence of correlations between the repeats (i.e. each single repeat is not equilibrated). Another possibility is to compare the results with a Markov Model analysis, such as TRAM (Wu et al., PNAS June 7, 2016 113 (23) E3221-E3230) or DHAM (Stelzl et al., J Chem Theory Comput. 2017 Dec 12;13(12):6328-6342) that should better handle non equilibrated samples.

We have now calculated autocorrelation functions of the interatomic distance used as a reaction coordinate in every restraint window. For the short distances, autocorrelations significantly larger than zero pertain for up to 10 ns. The autocorrelations from the short 2.5 ns simulations are significantly different from those obtained from longer 50 ns runs. Analyzing autocorrelations from the short trajectories may lead to a misleading interpretation that the simulation is readily converged, as the local free energy minimum is properly sampled and the trajectory appears to be memoryless. Unfortunately, this situation reveals the nature of the problem being akin to that of Zeno’s paradoxes: it is possible to deduce that the simulation is not converged only after obtaining a better converged simulation. The longer trajectories reveal that there are other relevant free energy minima, sampling which discloses longer autocorrelation times and subsequently has a pronounced effect on the free energy profiles.

“To further highlight the manifestation of undersampling, we have calculated autocorrelation functions of the interatomic distance used as a reaction coordinate in every restraint window (Figure 5D). […] The free energy profiles obtained from the trajectories of 10 ns (Figure 5C middle panel) already show that the average values for the three box sizes approach one another and the remaining differences are largely within the range of uncertainty.”

We have also performed DHAMed analysis, but in our hands incorporating the kinetic information had little effect on the generated free energy profiles when compared to those obtained by means of MBAR (Author response image 1). While this technique might help with certain convergence issues in particular situations, as illustrated in the DHAMed publication, we decided not to include this result in the manuscript, as it had no significant effect in the investigated case.

**Author response image 1. sa2fig1:** Comparison of the free energy profiles for dihydrofolate reductase as obtained by MBAR and DHAMed.

I report below more specific clarifications/improvements that need to be addressed.In the application to the Gb protein the authors compare their results to the work of Asthagiri and Tomar. Here the explicit comparison must be clarified; Asthagiri and Tomar also report a thermodynamic integration calculation to estimate the charging free energy of Gb which is (consistently with this work) box size independent. The box size dependence emerges when using the quasi chemical theory (e.g. to analyze the specific components). Additionally, the charging free energy reported by Asthagiri and Tomar doesn't seem consistent with the values reported in this work.

The reviewer correctly noticed that our results do not support those by Asthagiri and Tomar. We have now incorporated more details of the direct comparison to the work of Asthagiri and Tomar and provide explanation for the discrepancies.

“Overall, to obtain the net electrostatic contribution to solvation free energy we can use calculation in any of the sufficiently large solvated water boxes (Figure 2B) and subtracting the G value calculated in an infinitely large non-periodic vacuum box. This ensures that the electrostatic component of hydration free energy is independent of the box size. […] Considering different sampling times in this work and that reported by Asthagiri and Tomar might further contribute to the discrepancy in the estimated G_Q_ values.”

The Discussion section is more a result section about water diffusion, my suggestion is to move this section in the results and provide an actual discussion, e.g. expanding the Conclusions by including the considerations reported above.

The analysis of water diffusion and RDF has now been moved to the Results section forming a separate subsection on the solute kinetics and thermodynamics in the context of box size dependence.

Reviewer #2:General assessment:This paper aims to present a methodical demonstration of the importance of obtaining complete sampling statistics from MD simulations for the purpose of estimating thermodynamic or kinetic quantities. As a test case, the authors apply this demonstration to the hypothesis that solvent periodic box size affects the simulation-derived thermodynamics and kinetics of biomolecules. This work is statistically rigorous and provides a clear comparison between results obtained with insufficient sampling and those derived from more extensive sampling. This paper can serve as a good reference for future scientists employing MD simulations who want to make sure their sampling is adequate based on proper statistical methods.Major comments:1) The authors show that increased simulation time per window (50 ns instead of 2.5 ns) along the dihydrofolate reductase M20 loop motion reaction coordinate results in indistinguishable PMF profiles by box size. The authors attribute the discrepancies observed when using 2.5-ns sampling times to insufficient equilibration away from the initial structure, as 2.5 ns is not sufficiently greater than the autocorrelation time of the loop motion. While this is a very plausible explanation, the conclusion would be strengthened by explicit calculation of the reaction coordinate autocorrelation time. This is particularly important to include as an example to other scientists who would like to estimate how long their independent simulations need to be to avoid biased estimates that are caused by using the same initial structure.

We thank the reviewer for this suggestion. In fact, this comment resembles closely the third comment by the reviewer 1. To address both of these suggestions we have calculated autocorrelation functions and substantially expanded the Results section on the dihydrofolate reductase. A more detailed explanation is also already provided above.

2) For the kinetics analysis it would be helpful to include more information, including equations or a reference to the appropriate equations, to show how the rate constants and uncertainty were calculated in detail using the frequentist approach. In particular, it is not clear how the trials that did not exhibit any transitions for hemoglobin were accounted for in the calculation of the T state halflife.

We now added a description to the Materials and methods section.

“For hemoglobin simulations in each box size, we divided the overall pool of generated trajectories into smaller sub-samples, e.g. for the 9 nm box we randomly created sets of size 5, 10, 20 and 99. Within each of these sets we performed 1000 bootstrap runs, where for each run a survival curve was calculated (Gapsys and de Groot, 2019). Exponential fitting to a survival curve yields time constant, which can be related to the half-life by t_1=_2 = ln 2. Standard error was calculated as a standard deviation of a bootstrapped distribution. The described approach, however, does not allow estimating the half-life for those cases, where no transition is observed: for such situations, Bayesian method for rate estimation is to be used.”

Reviewer #3:I think this is a timely, important, and well-executed study that will be a valuable resource to the community. Insufficient sampling has been the bane of bimolecular simulation since its inception 40 years ago, and this paper is largely framed as a rebuke to a recent, high-profile example.My biggest complaint with the paper as it stands is not with the work, which is excellent, but with the lack of details in the presentation. As I see it, the real value in this manuscript isn't beating up on a bad paper, the authors already accomplished that, but in teaching the community how to do things better. The paper covers a truly enormous scope, arguably, 3-4 paper's worth, and as a consequence doesn't include all of the details needed to understand their arguments. Most of the information is available in the references, or in the code they distribute (another excellent choice on their part), but to really maximize the didactic value of the paper I think more of the explanation has to be in the main body of the paper. For example:

In the light of this suggestion, as well as the comments by the other reviewers, we have now substantially expanded all the sections of the manuscript, in particular providing more methodological details, as well as broadening the Results and Discussion sections.

1) They never said what enhanced sampling method was used in the DHFR calculations, or what the collective variable was. It sounds like umbrella sampling, but they weren't explicit.

A detailed description has now been added to the Materials and methods section, also a shortened version of the overall procedure is introduced in the Results section as well.

Results

“In the first example (Figure 5C upper panel), we constructed a profile from 3 umbrella sampling based simulation repeats running 2.5 ns, of which 0.5 ns were discarded for equilibration, for each of the 27 discrete windows along the reaction coordinate (distance between the C atoms of residues 18 and 45, as depicted in Figure 5B).”

Materials and methods:

“For the dihydrofolate reductase simulation setup, we closely followed the protocol described by Babu and Lim. Once solvated and neutralized, the systems were first equilibrated for 1 ns with position restraints on heavy protein atoms, followed by 1 ns unrestrained equilibration. The free energy profiles were constructed from equilibrium simulations at 27 discrete windows (umbrella sampling) along the reaction coordinate defined as a distance between the C atoms of residues 18 and 45. A harmonic restraint with a force constant of 4184 kJ/mol/nm^2^ was applied to the drive distance from 0.4 to 1.7 nm by a 0.05 nm increment. Three independent simulation repeats were performed for each of the three simulation boxes reaching 50 ns per window. The final PMF profile was calculated by means of MBAR estimator.”

2) Figures 8-10 on the kinetics of hemoglobin. There's no real explanation of what they did to generate any of these figures. The error bars in Figure 8 aren't defined, nor is how they were calculated. I'm assuming they did bootstrapping, with different sample sizes, but it should be stated explicitly. It's interesting that the standard errors in Figure 8B are larger for the smaller box, but the authors don't comment on why.

The procedure for obtaining error bars for Figure 8 has now been described in detail in the Materials and methods section (also addressed when replying to question 2 of reviewer 2). For the Figure 8B we have now added bootstrapped standard errors showing that the differences between the standard errors in the estimated half lives are statistically insignificant.

The theory behind Bayesian rate estimation has been added to the Materials and methods. In addition, we also describe describe in more detail the analysis performed for Figures 9-10.

“In the Bayesian framework, the rate estimates depend on the chosen prior distribution (more details in Materials and methods). Following Ensign and Pande, we chose both a uniform prior as well as Jeffreys prior for the rate estimates. From the generated hemoglobin trajectories, for the simulations in each box, we have extracted the number of transitions from T to R state, as well as the time required to make a transition (or the full trajectory time, in case no transition is made). This information allowed constructing posterior rate constant distributions (see Materials and methods and Ensign and Pande). As shown in Figure 9, whereas the choice of prior has a large influence on the probability distribution for the rate for single simulations, this effect is much smaller for 10, 20 and 100 repeats.”

3) Figure 9: there are 2 curves in each panel, presumably one for each prior, but which is which is not stated. In the top right panel, there's only 1 curve, with no explanation. Given that the paper is mostly intended as a teaching tool, I think a brief summary of the Ensign and Pande theory would be helpful, as would an explanation for why you would use to two different priors.

It is correct that the two curves denote posterior distributions calculated from the uniform and Jeffreys priors. This is denoted in the Figure 9 legend (top left panel). To make this clearer we have additionally added an explanation to the figure caption.

The details of Bayesian rate estimation are now provided in the Materials and methods section and the use of priors is discussed there as well.

Figure 9 caption:

“Two priors were used: uniform (solid line, subscript U) and Jeffreys prior (dashed line, subscript J).”

Materials and methods:

“The expressions above illustrate that the posterior distribution based on uniform prior allows infering rates even for the cases where no transitions have been observed. Jeffreys prior, although being invariant to scaling of , does not provide a defined posterior distribution in the absence of observed transitions. Therefore, using the uniform prior has an advantage that it allows making inferences about rates even when no transitions are observed. In practice, whenever possible, it is recommended to use both priors and compare the generated posteriors: given the sufficient evidence by the data, both posteriors converge to the same distribution.”

4) Figure 10: You have to read pretty far into the caption before you realize the data which shows 2 distributions is artificial, to prove a point. No details are given to explain how this was done, in the caption or the body of the paper. The authors should either leave it out or provide those details.

We have now indicated in the figure that the third panel in Figure 10 represents an artificial case. An additional explanation for this analysis was added to the text. We have also adjusted the synthetic case example, showcasing that to obtain strong evidence for two processes governing the kinetics of simulation in different boxes, one would need a very particular situation to occur. Namely, for three out of three 1 s hemoglobin simulations in a 15 nm box no transition should occur, while a 9 nm box in three simulations would make a transition already in the first half of the simulation time.

“The Bayes formalism provides a framework to estimate if two distributions differ significantly from each other (details in Materials and methods section). […] To reach the scenario of strong evidence for two disparate processes generating the distributions, we had to artificially replicate 3 times the observations that in 9 nm box a transition occurs in 0.466 s and no transition occurs in 15 nm box within 1𝜇s.”

5) Very few details are given about the alanine dipeptide calculations.

More details about the alanine dipeptide simulations have been added to the Materials and methods.

“To summarize briefly, we have prepared four simulation setups in cubic boxes with an edge of 3.0, 5.0, 7.0 and 9.0 nm. […] The standard errors were calculated from the standard deviations of independent simulations.”

6) When discussing the protein solvation calculations, the final conclusion (that the appropriate approach to eliminate box-size dependence in vacuum is to use no cutoff at all) is very reasonable, but the data should be shown.

This approach is demonstrated in Figure 2 where infinitely large box (no periodicity) was used to calculate G in vacuum. We have now clarified this in the text.

“Overall, to obtain the net electrostatic contribution to solvation free energy we can use calculation in any of the sufficiently large solvated water boxes (Figure 2B left panel) and subtract the G value calculated in an infinitely large non-periodic vacuum box (illustrated by the blue square in Figure 2B middle panel).”